# PROSODYLM: Uncovering the Emerging Prosody Processing Capabilities in Speech Language Models

**Kaizhi Qian**[1,2], **Xulin Fan**[3], **Junrui Ni**[3], **Slava Shechtman**[1], **Mark Hasegawa-Johnson**[3], **Chuang Gan**[1,2,4], **Yang Zhang**[1,2]

[1] IBM Research     [2] MIT-IBM Watson AI Lab, USA
[3] University of Illinois at Urbana-Champaign, USA     [4] UMass Amherst, USA
{kqian, chuangg, yang.zhang2}@ibm.com,
{xulinf2, junruin2, jhasegaw}@illinois.edu, slava@il.ibm.com

## Abstract

Speech language models refer to language models with speech processing and understanding capabilities. One key desirable capability for speech language models is the ability to capture the intricate interdependency between content and prosody. The existing mainstream paradigm of training speech language models, which converts speech into discrete tokens before feeding them into LLMs, is sub-optimal in learning prosody information — we find that the resulting LLMs do not exhibit obvious emerging prosody processing capabilities via pre-training alone. To overcome this, we propose PROSODYLM, which introduces a simple tokenization scheme amenable to learning prosody. Each speech utterance is first transcribed into text, followed by a sequence of word-level prosody tokens. Compared with conventional speech tokenization schemes, the proposed tokenization scheme retains more complete prosody information, and is more understandable to text-based LLMs. We find that PROSODYLM can learn surprisingly diverse emerging prosody processing capabilities through pre-training alone, ranging from harnessing the prosody nuances in generated speech, such as contrastive focus, understanding emotion and stress in an utterance, to maintaining prosody consistency in long contexts.

## 1 Introduction

As large language models (LLMs) are becoming increasingly adept in communicating with humans and responding to human requests, there is an increasing need to develop language models with speech capabilities. The easiest way is to append an automatic speech recognition (ASR) module to convert speech to the input text to an LLM, and a text-to-speech (TTS) module to convert the LLMs' output text back into speech. However, such a paradigm loses the rich information in speech beyond text, and thus the resulting LLM is unable to understand the nuances in speech communication with humans.

To understand what capabilities a speech LM should have, consider two examples. In the first example, suppose an LLM, as an agent, says, '*Today's weather is sunny.*' Then a human, as a user, says, '*I am sorry. Can you say it again?*' Then, the LLM repeats, '*Today's weather is sunny.*' In a nautural speech conversation, the repeated utterance should be slower and more enunciated than the first utterance. However, a TTS, typically trained without a sophisticated understanding of the context, may just produce a similar repeated utterance.

As the second example, suppose the LLM agent asks '*How are you today?*', and the human user, despite feeling down and upset, lies, '*I am feeling good.*' In a natural speech conversation, the agent may sense the user's upset tone, and may ask the user if there is anything wrong, or at least adjust to a more comforting intonation. In the above paradigm, however, the LLM could not notice the upset emotion from the text response.

The above examples underscore a crucial capability that a speech-augmented LLM must possess — the ability to *capture the intricate interdependency between content and prosody*.

Content and prosody are two fundamental aspects of speech. While content is typically represented in text transcriptions and well-handled by LLMs, prosody remains unfamiliar to them. Consequently, effective modeling of prosody and its connection to content is essential for training speech-augmented LLMs. Specifically, three key dependencies must be learned:

• **Content→Prosody:** How preceding content influences the prosody of generated speech., *e.g.*, slowing down appropriately when asked to repeat something.

• **Prosody→Content:** How prior prosody affects subsequent content generation, *e.g.*, asking if there is anything wrong in response to an upset tone from the user. As an alternative example, a sentence uttered in a rising tone (suggesting a question) should be answered differently than when spoken in a falling tone (suggesting a declarative statement).

• **Prosody→Prosody:** How previous prosody shapes the prosody of the next utterance, *e.g.*, adopting a comforting tone in response to a user's distressed prosody.

To address the prosody modeling problem, the mainstream approach to training speech LLM would tokenize the speech signal, and then pre-train the LLM on the speech tokens. Since the speech tokens would contain some prosody information, such a paradigm should ideally teach the LLM to capture the aforementioned prosody dependencies. Unfortunately, we find that simply pre-training on the speech tokens alone is not sufficient to enable LLMs to learn much prosody information, if at all, even when the pre-training data reaches 30k hours. In fact, many existing works have to rely on sophisticated fine-tuning or instruction-tuning to force speech LLM to produce or understand prosody information appropriately (Huang et al., 2025), which may only work for highly specified prosody processing capabilities. These observations lead us to the following research question: Is there a way to enable the LLM to **develop diverse prosody modeling abilities as emerging capabilities through pre-training only**, just like how LLMs acquire other emergent skills in the text domain?

We hypothesize that the inefficacy in learning prosody is due to the design of most speech tokenization schemes not optimized for prosody learning. First, many speech tokenization schemes prioritize capturing the semantic information, not prosody, and thus the resulting speech tokens may already lose a lot of prosody information. Second, most speech tokens may be too abstract to the LLM, and it may take an LLM a significant portion of its representation power just to re-align the speech tokens. In short, the crux to the problem may just be designing a tokenization scheme that better retains the prosody information and that is more understandable to LLMs.

Motivated by this, we proposed PROSODYLM, a speech LLM that built upon a very simple tokenization scheme — each speech utterance is first transcribed into text, followed by a sequence of word-level prosody tokens, describing the F0, duration, and energy behaviors for each word. Since each dimension of the prosody token has a very straightforward high and low interpretation, and the content information is already disentangled and converted to text, this tokenization scheme should make it much easier for the LLM to learn prosody. Finally, to convert prosody tokens from and to speech waveform, we adapt the encoder and decoder in StyleTTS2 (Li et al., 2023), which we find can work seamlessly with PROSODYLM and generate vivid and natural-sounding speech.

Simply pre-trained on 30k hours of audiobooks in Librilight (Kahn et al., 2020), PROSODYLM demonstrates superior prosody understanding capabilities, across all three aforementioned dependency categories. PROSODYLM can harness the prosody nuances, such as contrastive focus, in appropriate settings, correctly recognize emotion and stress in speech utterances, and maintain prosody consistency in long contexts, all *without* being trained to perform the tasks. The findings of this paper, as well as the comprehensive prosody evaluation schemes, can contribute to more expressive and natural human-AI interactions.

## 2   Related Work

**Speech Tokenization.**    Recent work has advanced the representation of speech as discrete tokens for language modeling. Borsos et al. (2023a) introduce a two-level tokenization into semantic and acoustic tokens, while Borsos et al. (2023b) predict SoundStream codec tokens

non-autoregressively. Zhang et al. (2023c) disentangle speech attributes hierarchically across RVQ layers, and Zhang et al. (2023b) propose SpeechGPT-Gen for cascaded generation of semantic and prosodic tokens. LAST (Turetzky & Adi, 2024) aligns speech tokens with language models using LM-aware training. DM-Codec (Ahasan et al., 2024) distills multimodal information into unified tokens, and RepCodec (Huang et al., 2023) reconstructs SSL features for quantization. dMel (Bai et al., 2024) discretizes mel-filterbank intensities, while WavTokenizer (Ji et al., 2024) achieves efficient token compression. DC-Spin (Chang et al., 2024) learns speaker-invariant, phonetic-rich tokens. A recent survey by Guo et al. (2025) reviews these trends. PROSODYLM builds on this literature with a prosody-driven tokenization scheme that enhances prosody modeling.

**Speech Language Models**    Numerous spoken language models (SLLMs) have emerged for direct speech-based interaction. Early efforts like AudioLM demonstrated coherent prediction of audio tokens (Borsos et al., 2023a), followed by GSLM and dGSLM for unsupervised speech and dialogue modeling (Lakhotia et al., 2021; Nguyen et al., 2023). SpeechGPT (Zhang et al., 2023a) and SpeechGPT-Gen (Zhang et al., 2023b) introduced cascaded generation of semantic and prosodic tokens for controllable speech synthesis. Later models like SpiritLM (Nguyen et al., 2025), USDM (Kim et al., 2024), and SpeechSSM (Park et al., 2024) introduced multimodal reasoning, prosody awareness, and long-form speech generation. Recent systems expand to full dialogue: AudioPaLM supports speech-to-speech translation (Rubenstein et al., 2023), GLM-4-Voice enables bilingual voice interaction (Zeng et al., 2024a), and Moshi provides full-duplex real-time communication with Mimi codec (Défossez et al., 2024). WHISMA (Li et al., 2024) improves zero-shot speech understanding, while Mini-Omni (Xie & Wu, 2024) and Freeze-Omni (Wang et al., 2024) offer low-latency, streaming speech interaction. These models largely overlook systematic evaluation of prosody processing. Our work fills this gap.

**Expressive Speech Synthesis**    Expressive text-to-speech (TTS) systems have rapidly improved in prosodic quality. Commercial systems like ElevenLabs (ElevenLabs, 2023) and Sesame (Iribe et al., 2025) now set high benchmarks for realism. In research, models have explored various ways to control prosody. Reference-based approaches such as VALL-E (Wang et al., 2023) and NaturalSpeech 2 (Shen et al., 2023) use short recordings or prompts to synthesize speech with matching speaker and intonation in zero-shot settings. Other methods learn latent style representations: mixture-of-experts architectures specialize on distinct speaking styles (Jawaid et al., 2024), while CLAPSpeech (Ye et al., 2023) uses contrastive pretraining to model prosody from text context. Diffusion models like DrawSpeech (Chen et al., 2025) enable fine-grained prosody control through user-drawn sketches. ProsoSpeech (Ren et al., 2022) leverages quantized vector pretraining to enhance expressivity, and DC-TTS variants (Chang et al., 2024) explore speaker-invariant prosody modeling.

## 3  PROSODYLM

PROSODYLM is a speech language model with superior prosodic modeling capabilities. The key design idea is to tokenize the speech in a way that is more amenable to prosody modeling. As shown in Figure 1, PROSODYLM consists of three modules:

• A **speech encoder**, which encodes speech in to a discrete token sequence;

• A **language model** (LM), which, given historical speech token sequence as context, generates the future speech token sequence in an auto-regressive manner;

• A **speech decoder**, which converts the speech token sequence into speech waveform.

We will first discuss the speech encoding and tokenization scheme of PROSODYLM in Section 3.1; and then introduce the encoder and decoder in Section 3.2. Finally, we will discuss different generation modes of PROSODYLM in Section 3.4.

### 3.1  Speech Tokenization

In PROSODYLM, the speech utterance is tokenized into a sequence with explicit prosody information, called *prosody tokens*. Each speech utterance is divided into sentences, and

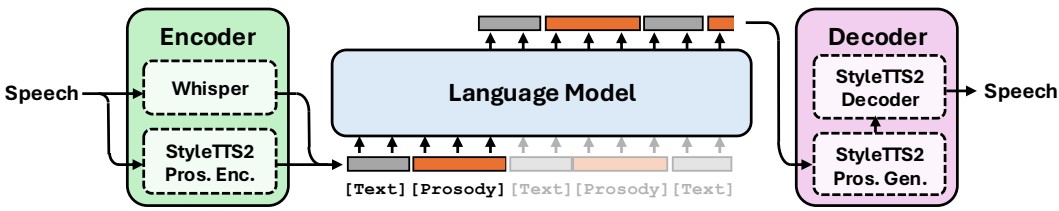

Figure 1: The overall framework of PROSODYLM.

each sentence is encoded into two sections, the [Text] section followed by the [Prosody] section, separated by two special tokens, <SEP1> and <SEP2>, respectively. The [Text] section contains the text transcription of the utterance, *e.g.*, 'How are you?' He asked.

The [Prosody] section defines word-level prosody in the following format:

```
[Global] <SIL> [Dur] how [5-dim Prosody] <SIL> [Dur] are [5-dim Prosody] ...
```

As can observed, the prosody sequence starts with a [Global] token, then alternates between two patterns, ❶ <SIL> [Dur], which defines the pause length between consecutive words, and ❷ [word] [5-dim prosody], which defines five-dimensional prosody vector for each word, introduced by Shechtman (2023). The five dimensions are:

1. **Duration:** The average duration (in frames) of the phones in the word;

2. **Log-F0 range:** The the difference between the 95% and 5% percentiles of the frame-level Log-F0 contour of the word;

3. **Log-F0 median:** The median of the frame-level Log-F0 contour of the word;

4. **Log-F0 slope:** The slope of the straight line that best fits the Log-F0 contour of the word;

5. **Log-energy:** The log-norm of the mel-spectrogram of the word.

Each dimension of the prosody vector is clipped above and below (thus the value range becomes a finite interval), then normalized into the interval $[0, 1]$, and finally quantized into 512 equally-spaced bins as the prosody tokens. All five dimensions share the same 512-sized vocabulary, added to the LLM's vocabulary. Further details are provided in Appendix A.

The optional [global] token defines sentence-level prosodic characteristics. In our implementation, we use the *extremity score*. For each prosody token in a sentence, we compute its occurrence frequency in the pre-training set. These frequencies are then aggregated across all prosody tokens in the sentence to obtain the extremity score. A lower score indicates that the sentence contains rarer prosody tokens, and thus the resulting prosody is likely to sound more extreme. The extremity score is discretized into a 512-sized vocabulary, using the same tokenization method as for other prosody tokens. Empirically, we find that the [global] token is better for generation tasks, whereas removing it is better for understanding tasks.

### 3.2 Speech Encoder and Decoder

The speech encoder and decoder both borrow components of STYLETTS2 (Li et al., 2023), which is a TTS system with competitive sound quality.

**Speech Encoder.** The speech encoder needs to extract both the [Text] and [Prosody] sections of the tokenized sequence from input speech. For [Text], we adopt the Whisper (Radford et al., 2023) to derive the transcriptions. For [Prosody], we use the feature extractor components in STYLETTS2 to compute the five-dimensional prosody vector. Specifically, to extract duration, we use the text aligner in STYLETTS2 to extract the phone-level boundaries (in frame), from which the phone durations are computed. To extract the log-F0 contour and energy, we directly adopt the inherent pitch extractor and energy extractor in STYLETTS2, respectively. All the encoder components are adopted as is without any modifications.

**Speech Decoder.** We adapt the synthesis modules in STYLETTS2 to convert the output speech token sequences into audio waveforms. The original STYLETTS2 needs to generate the prosody information all on its own. In our pipeline, however, the word-level prosody

information is already generated by the preceding LM, so the decoder only needs to convert the word-level prosody information to finer-grained. Therefore, we need to modify the prosody generation modules in STYLETTS2.

Specifically, the original STYLETTS2 has a duration predictor and a prosody predictor, which, given a style embedding and input phone embeddings, produce phone-level duration and frame-level F0 and energy, respectively. We replace their conditioning on the style embedding with that on our predicted word-level duration, F0 and energy, respectively. More implementation details about the speech decoder can be found in Appendix B.

### 3.3 Pre-training Details

As discussed in Section 1, we are interested in eliciting the prosody capabilities via pre-training only. To pre-train PROSODYLM, we initialize the model with Llama3.1-8b-Instruct[1], then continuously pre-train it to perform next-token prediction on 29.9k hours of audiobooks from Librilight (Kahn et al., 2020), tokenized into the format specified in Section 3.1, using rank-64 LoRA (Hu et al., 2022) with $\alpha = 16$.

We performed data cleaning on the extracted prosodic features based on the following criteria. To balance the amount of data per speaker, speakers with a total speech duration of less than one hour were excluded entirely. Additionally, due to poor audio quality or other factors, some utterances yielded extreme or invalid values during prosody extraction. If more than 20% of the prosodic tokens in an utterance were extreme or invalid, the utterance and its corresponding audio were removed from the training set.

We append a simple instruction before the tokenized sequence, which is 'Spin a narrative [SPK F0]'. The [SPK F0] refers to the average log-F0 of the speaker, normalized and tokenized the same way as the second dimension of the five-dimensional prosody token vector (see Section 3.1). This is very important for accurate prediction because typical male and female speakers have very different pitch ranges. Note that this is the only speaker-dependent information accessed by LM.

To avoid overfitting to a specific instruction, we also prompt ChatGPT-4o to generate 65 paraphrases of 'Spin a narrative', which will be randomly selected as the instruction for each sequence during pre-training. Some of these instructions are listed in Appendix C.

### 3.4 Generation Modes of PROSODYLM

Since its speech sequence is arranged into the '[Text][Prosody][Text][Prosody]...' pattern, PROSODYLM can be used for multiple speech generation tasks.

**TTS.** PROSODYLM can synthesize multi-sentence utterances from text as follows, by filling the text in to the [Text] section, and then have the LM generate the [Prosody] section.

**Speech Continuation.** Given a reference speech utterance, we can generate the continuation by feeding the tokenized reference speech to the LLM as the context.

## 4 Experiments

In this section, we will explore how PROSODYLM, only pre-trained on around 30k hours of audiobooks (details in Appendix C), can capture the prosodic nuances in different speech generation tasks. As discussed in Section 1, there are different types of prosody dependencies, *Content→Prosody*, *Prosody→Content*, and *Prosody→Prosody*, we will design test cases to cover all three types, which is discussed in Sections 4.2 to 4.4, respectively. PROSODYLM is pre-trained on audiobooks only, the test cases are all designed to follow the audiobook style. We will use global control for generation tasks (Section 4.2) and no global control for understanding tasks (Sections 4.3 and 4.4). An ablation study on global control is provided in Appendix E.3. We encourage readers to listen to our audio demos[2].

---

[1] https://huggingface.co/meta-llama/Llama-3.1-8B-Instruct
[2] Demo and code webpage: https://auspicious3000.github.io/ProsodyLM-Demo/

### 4.1 Baselines

We introduce two groups of baselines. **Group-A baselines**, containing models that are almost the same as PROSODYLM (in terms of training data, pre-training settings, *etc.*), except that the prosody tokens (in the [Prosody] section) are replaced with other speech tokens:

• STYLETTS2 (Li et al., 2023): The original STYLETTS2, not connected to LMs.

• MIMI-token (Zeghidour et al., 2021; Défossez et al., 2022): Our method using MIMI tokenization scheme, used in the MOSHI speech language model Défossez et al. (2024). MIMI encodes 24kHz speech into 8 tokens per 12.5ms frame, with the first seven dimensions modeling the acoustics in speech, and the last dimension semantics.

• GLM4V-token (Zeng et al., 2024b): Our method using the tokenization scheme in GLM-4-Voice (Zeng et al., 2024a), producing 12.5-Hz tokens from whisper-large-v3 representations.

**Group-B baselines** consists of state-of-the-art commercial or open-sourced pre-trained speech generation models, including:

• STEP-AUDIO-TTS-3B (Huang et al., 2025): A state-of-the-art speech generative model with language modeling support.

• ELEVENLABS (ElevenLabs, 2023): A state-of-the-art commercial expressive TTS system.

• GPT-4O[3]: One of the best multi-modal LLM with superior speech generation capabilities.

The language models of group-A baselines are all **only pre-trained on 29.9k hours of audiobooks** from Librilight (Kahn et al., 2020), without any fine-tuning or instruction-tuning. More pre-training details can be found in Appendix C. In contrast, group-B baselines has significant unfair advantages, because they are trained on much more and higher-quality data, and have undergone various tuning and alignment to improve the quality. Nevertheless, we introduce these baselines as reference points to gauge the strength of the emerging prosody processing capabilities despite the simple training scheme of PROSODYLM.

PROSODYLM, STYLETTS2, MIMI-tok, and STEP-AUDIO-TTS-3B support voice cloning, so we use the same set of voices, which are randomly drawn from Librilight, for all the experiments. For the other baselines, we use the voices that they support. Further experiment details can be found in Appendix D.1, baseline details in Appendix D.2.

### 4.2 Content→Prosody

Content→Prosody refers to the capabilities of generating prosody that fits the content. In this section, we present three tasks, from easy to hard, to test such capabilities.

#### 4.2.1 Direct Style Specification

In audiobooks, it is very common to have quotes of a character followed by a style, *e.g.*,

> "Time moves forward, whether we're ready or not!" He said in a high voice.

We would like to test whether PROSODYLM can vary the prosody of the quote according to the style specification. To this end, we introduce three pairs of styles, which are ❶ an F0 pair: '*in a high voice*' v.s. '*in a low voice*', ❷ a duration pair: '*quickly*' v.s. '*slowly*', and ❸ an energy pair: '*loudly*' v.s. '*quietly*'. We also prompt ChatGPT-4o to generate 10 neutral quotes, which, concatenated with the 6 styles, form 60 scripts, each synthesized in 20 different voices.

For each F0 pair with the same quote and speaker, we calculate the difference in average F0 between the two utterances in this pair (*high voice* minus *low voice*), which are then aggregated across all F0 pairs. Similarly, we compute the average difference in symbol rate for the duration pair (*quick* minus *slow*), and the average difference in energy for the energy pair (*loud* minus *quiet*). If the model follows the style specification, the differences would be positive. Further details can be found in Appendix D.3.1.

---

[3]https://openai.com/index/hello-gpt-4o/

| Methods | Direct Style Specification | | | Indirect Style Inference | | |
|---------|-------|-------|-------|-------|-------|-------|
| | F0 pair | Dur. pair | Energy pair | F0 pair | Dur. pair | Energy pair |
| STYLETTS2 | 3.25 (1.77) | 0.28 (0.16) | 0.125 (0.092) | 0.80 (1.41) | 0.03 (0.13) | **0.152** (0.062) |
| MIMI-tok | 1.86 (1.67) | 0.42 (0.21) | 0.126 (0.091) | 5.84 (2.26) | 0.28 (0.19) | 0.104 (0.055) |
| GLM4V-tok | 5.53 (1.85) | 0.54 (0.26) | 0.037 (0.084) | -3.38 (1.83) | 0.83 (0.13) | 0.081 (0.037) |
| PROSODYLM | **18.51** (1.79) | **1.48** (0.17) | **0.171** (0.058) | **21.88** (1.15) | **1.17** (0.14) | 0.061 (0.016) |
| STEP-AUDIO | 3.18 (1.52) | 1.19 (0.17) | 0.258 (0.075) | 5.42 (2.39) | 1.77 (0.64) | 0.315 (0.067) |
| ELEVENLABS | 7.04 (2.54) | 0.11 (0.18) | 0.165 (0.111) | 17.95 (3.03) | -0.26 (0.17) | 0.044 (0.025) |
| GPT-4O | 115.36 (5.77) | 4.36 (0.28) | 1.143 (0.141) | 30.57 (3.85) | 1.88 (0.20) | 0.302 (0.064) |

Table 1: Results of direct (Section 4.2.1) and indirect style following (Section 4.2.2). Numbers in parentheses are standard deviations. Best performance within group A is bolded.

Table 1 (left) shows the prosodic differences in each style pair. PROSODYLM achieves a large positive value in all three style pairs, outperforming most baselines, even group-B, except for GPT-4O, which is clear evidence that it can follow direct style specifications.

### 4.2.2 Indirect Style Inference

Moving on to a slightly more challenging setting, we study ability to generate expressive quotes based on indirect descriptions of characters. Here is an example.

> The little fox twitched its ears and bounced on its paws, its golden fur standing on end with excitement. Every movement was sharp and sudden . . .
> "Time moves forward, whether we're ready or not!" She said.

Although the paragraph does not specify how the fox speaks, it can be deduced that she is likely to speak in a light, high-pitched voice. We would like to test whether PROSODYLM can make such an inference. We prompt GPT-4o to generate descriptions of 6 characters, each likely associated with one of the aforementioned 6 styles. These descriptions have been verified as indicative of the corresponding styles by a human evaluation (see Appendix E.1). We then combine it with the same 10 quotes as in Section 4.2.1. The speakers and the objective metrics are also the same as in Section 4.2.1. Further details can be found in Appendix D.3.2.

Table 1 (right) shows prosodic differences, which are generally smaller than direct style specification, reflecting increased difficulty of the task. Nevertheless, PROSODYLM still exhibits similar competitiveness, outperforming almost all the baselines except for energy.

### 4.2.3 Clarification

We introduce an even more challenging setting in which prosody must reflect contextual dependencies that span multiple sentences. Consider the following paragraph:

> "Isabella pushed the chair forcefully," Tom said.
> "Did you say Isabella dragged the chair forcefully?" Jerry asked, apparently not paying attention.
> "No, I said Isabella pushed the chair forcefully!"

In such clarification settings, humans would typically emphasize the word '*pushed*' in the final utterance, a well-known prosody phenomenon named **contrastive focus** (Chafe, 1976).

To test whether PROSODYLM learns contrastive focus, we prompt GPT-4O to generate 10 simple sentences, each consisting of a subject, a verb, an object, and an adverbial phrase. For each sentence, we then prompt GPT-4O to generate four misunderstandings—one for each sentence component—resulting in a total of 40 passages (more details in Appendix D.3.3).

For each component in the final quote, say *verb* for instance, we compute its average F0. We then aggregate the average F0 across the final-quote-verbs in all the utterances under three cases: **pre-focus**, **on-focus**, and **post-focus** cases, where the verb is before, at, or after the component that should be emphasized, respectively.

Figure 2: Average F0 under different focus conditions for selected methods.

| | Consistent Voices | | | Inconsistent Voices | |
|---|---|---|---|---|---|
| **Methods** | Prosody | Naturalness | **Methods** | Prosody | Naturalness |
| STYLETTS2 (A) | 3.51 (0.08) | 3.80 (0.07) | GLM4V-tok (A) | 3.86 (0.07) | 3.87 (0.07) |
| MIMI-tok (A) | 2.82 (0.07) | 2.34 (0.08) | ELEVENLABS (B) | **4.08** (0.07) | **4.23** (0.06) |
| STEP-AUDIO-tok (B) | **4.01** (0.07) | **4.04** (0.06) | GPT-4O (B) | **4.05** (0.07) | 3.75 (0.08) |
| PROSODYLM | **4.07** (0.06) | **3.96** (0.06) | PROSODYLM | **4.00** (0.07) | 3.90 (0.07) |

Table 2: Subjective MOS evaluation results. Numbers in parentheses are standard deviations. The scores that do not significantly differ ($p > 0.01$) from the best scores in a column are denoted in **bold**. (A) and (B) indicate baseline groups.

Figure 2 shows average F0 across the three cases for selected methods (full results shown in Appendix E.2), revealing that PROSODYLM learns both features of contrastive focus: ❶ *On-focus stress*: the on-focus case always has the highest F0, and ❷ *Post-focus compression*: the F0 in the post-focus case is always significantly suppressed. On the contrary, few other baselines capture these characteristics, except for GPT-4O.

Additional experiments in Appendix E.2 show that PROSODYLM also learns to **slow down** in the final quote, another prominent prosodic behavior for clarification.

### 4.2.4 Subjective Test

To evaluate whether the improved prosody modeling of PROSODYLM enhances perceptual quality, we conduct two distinct Mean Opinion Score (MOS) listening tests on Amazon Mechanical Turk[4], one on systems that can generate the same voices as PROSODYLM (called consistent voices), and the other on systems that cannot (called inconsistent voices). 36 audiobook excerpts (4-8 sentences each, outside of PROSODYLM's pre-training data) are selected using GPT-4o and synthesized using four voices (two male, two female) for all systems but GLM4V-token that has just a single female voice available. Participants are asked to rate each sample on two 1–5 scales: one for overall perceptual quality and one for prosody quality, i.e., how well the intonation fits the context.

The results of the subjective tests are presented in Table 2. They reveal that PROSODYLM significantly outperforms all the group-A baselines in both prosody quality and naturalness, and performs comparably to all group-B baselines featuring professional speakers and recording environments, particularly in terms of prosody quality. This verifies the superior prosody modeling of the prosody tokens. More details can be found in Appendix D.3.4.

### 4.3 Prosody→Content

Prosody→Content refers to the ability to capture how prosody in prior speech influences subsequent text generation. We introduce two experiments to study such an ability.

### 4.3.1 Emphasis Detection

Emphasis detection refers to the task of determining which words in a reference speech are emphasized. For PROSODYLM, pre-trained solely on audiobooks where emphasis labels are sparse, it would be surprising if the model has learned the concept of emphasis at all.

---

[4]https://www.mturk.com/

| Methods | Happy | Sad | Excited | Angry | Neutral |
|---|---|---|---|---|---|
| MIMI-tok | -0.010 (0.005) | 0.154 (0.009) | 0.008 (0.006) | -0.013 (0.006) | -0.009 (0.007) |
| GLM4V-tok | -0.014 (0.005) | 0.030 (0.007) | -0.010 (0.005) | 0.002 (0.005) | 0.001 (0.006) |
| PROSODYLM | **0.014** (0.007) | **0.203** (0.015) | **0.203** (0.017) | **0.021** (0.010) | **0.063** (0.014) |

Table 3: Average increase in log output probability emotion recognition. Numbers in parentheses are standard deviations.

To test this, we collect a set of reference utterances from EmphAssess (de Seyssel et al., 2023), consisting of parallel recordings of the same content by the same speaker, but with emphasis placed on different words. We construct speech sequences of the following form:

> "And that to me was so exhilarating." [Prosody (Ref Speech)] He/She emphasized '____'.

For each word in the quote, we compute the difference in the log-probability of the model selecting that word in the blank position, comparing the case where the reference speech emphasizes the word versus when it does not. We then aggregate these differences across all words and reference utterances. If PROSODYLM could understand emphasis, the aggregated difference should be significantly positive. Formally, let $q(W|u)$ denote the model's probability of generating word $W$, conditioned on reference speech $u$. Let $\mathcal{U}$ denote a set of parallel utterances (same content & speaker, different emphasis). Our metric is defined as:

$$\mathbb{E}_{\mathcal{U},w}\left[\mathbb{E}_{\{u\in\mathcal{U}:w \text{ is emphasized}\}}\log q(W=w|u) - \mathbb{E}_{\{u\in\mathcal{U}:w \text{ is not emphasized}\}}\log q(W=w|u)\right]. \tag{1}$$

More details can be found in Appendix D.4.1. The reason why we measure the relative log-probability differences instead of directly measuring whether each synthesized word is emphasized is that one major complication of this experiment is that the emphasis depends not only on the prosody of the quote, but also on the content of the quote. Since the datasets are designed to be parallel (same content paired with different emotions/emphasis words), the prosody cues and content cues are often in conflict. Take the above quote as an example, the content of the quote would strongly suggest 'exhilarating' should be emphasized, which creates a significant bias towards certain words, even if these words were not prosodically emphasized. Only by computing the relative log-probability differences can we separate the effect brought by prosody changes, controlling for the influence of content.

Table 4 (left) lists the results. Only two group-A baselines are included here because it is infeasible to fit the rest into the Prosody→Text generation mode. The results show that only PROSODYLM displays a statistically significant increase in the probability (6.6%), while the baselines can barely recognize emphasis.

### 4.3.2 Emotion Recognition

We now turn to the more challenging task of recognizing emotions from a reference utterance. We adopt the JL corpus (James et al., 2018), which contains five emotion categories — *happy*, *sad*, *excited*, *angry*, and *neutral*. For each category, we select all parallel reference utterances with the same script and speakers but different emotions, and construct speech sequences of the following form:

> "John laughs like your father." [Prosody (Ref Speech)] He/She was ____.

Similar to Equation (1), for each emotion word ('*happy*', '*sad*', '*angry*', '*excited*', '*neutral*'), we compute the increase in log-probability when the ground-truth emotion of the reference utterance matches the word versus when it does not. We aggregate the differences separately for each output emotion word. More details can be found in Appendix D.4.2.

The results in Table 3 show that PROSODYLM can effectively detect all five emotions, especially for *Sad* and *Excited*. The baselines show almost no significant probability increase.

| Methods | Emphasis Detection | Dialogue | | |
|---|---|---|---|---|
| | | F0 Pair | Dur. Pair | Energy Pair |
| MIMI-tok | 0.0044 (0.0043) | 0.73 (2.46) | -0.61 (0.25) | -0.147 (0.098) |
| GLM4V-tok | 0.0073 (0.0037) | 1.56 (1.09) | 0.37 (0.16) | -0.052 (0.048) |
| PROSODYLM | **0.0656** (0.0062) | **47.9** (3.6) | **1.09** (0.14) | **0.038** (0.047) |

Table 4: Emphasis Detection (left) and Dialogue continuation (right) results. Numbers in parentheses are standard deviations.

## 4.4 Prosody→Prosody

Finally, we study whether PROSODYLM can also vary the prosody of the subsequent generation according to the preceding prosody. Consider the following conversation:

> "Some things are understood without being spoken!" [Prosody (Ref Speech 1)] Tom said.
> "Not every question has a simple answer!" [Prosody (Ref Speech 2)] Mary said.
> "The past is always part of the present!" Tom said.
> "Time moves forward, whether we're ready or not!" Mary said.

There are two rounds of conversation between Tom and Mary. In the first round, a reference speech is provided for each speaker. In the second round, we let the model synthesize the speech given the text. If the styles in the two reference speech utterances are different, we expect the continuation of the conversation to alternate between the two different styles. A common mistake of existing methods is that they tend to maintain the same style across different speakers, ignoring the character change.

We select three pairs of reference utterances synthesized by PROSODYLM in Section 4.2.1 for each speaker. The first pair exhibits the greatest contrast in average F0, the second in average symbol rate, and the third in energy. For each speaker and reference pair set as the first-round dialogue, we select two distinct quotes from the 10 quotes used in Section 4.2.1 as the script for the second-round dialogue and let the model synthesize them.

We then measure whether the prosodic contrast in the reference utterances is carried over to the second-round dialogues. For reference pairs with the F0 contrast, we compute the F0 difference between the two utterances in each second-round dialogue and aggregate them. Similarly, for symbol rate and energy reference pairs, we compute the corresponding aggregated symbol rate and energy differences, respectively.

Table 4 (right) shows the results, where PROSODYLM exhibits significant prosody contrast between the two speakers, except for the energy pair, while the baselines show almost no contrast. This shows the strong style copying capability of PROSODYLM.

## 4.5 Content→Content

Although PROSODYLM is designed for prosody related task, we would like to investigate how much PROSODYLM retains the text processing capabilities. Please refer to Appendix E.4 for the additional experiments.

## 5 Conclusion

In this paper, we uncover a surprisingly wide range of emerging prosody processing capability in a pre-trained speech LM. By explicitly tokenizing the prosody information and content, the resulting LM can generate very expressive speech, develop a preliminary understanding of emphasis and emotion, and can clone the styles in reference speech, which inspires a promising paradigm of speech LM training. Still, our work has some limitations. First, our prosody token has a limited expressive power compared with other speech tokens, for it cannot capture changes in voice quality. Second, most parameters in our decoder are fixed to the StyleTTS pre-trained weights, and thus the sound quality might be compromised. We will tackle these problems as future directions.

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

## A  Speech Tokenization Details

The details of computing the prosody tokens are discussed below.

**Duration.**   After phonemizing the text, we obtained the alignment between the phoneme sequence and the spectrogram frames using the alignment module of StyleTTS2. This allowed us to determine how many spectrogram frames correspond to each symbol in the phoneme sequence. Given the start and end indices of each word, we then calculated the average number of frames per symbol for each word. Finally, we took the logarithm of this value to obtain the duration information for each word.

**Log-F0.**   The F0-related features include three quantities: F0 range, F0 median, and F0 slope. We first input the spectrogram of the entire utterance into the F0 extraction module of StyleTTS2 to obtain the frame-level F0 values, and then apply a logarithmic transformation. Using the alignment and word boundary information obtained during the duration extraction process, we derive the F0 contour for each word. The F0 range is computed as the difference between the 95th and 5th percentiles of the contour, the F0 median is given by the median value, and the F0 slope is computed using linear regression via the `linregress` function from SciPy.

**Energy.**   We computed the spectrogram using the `MelSpectrogram` function from torchaudio, with a window length of 1200 samples and a hop length of 300 samples, which are consistent with those used by StyleTTS2. The same spectrogram was used for extracting the duration and F0 features described above. For energy computation, we first calculated the L2-norm of each frame, followed by a logarithmic transformation to obtain the log-norm of the spectrogram. Using the alignment and word boundary information obtained during duration extraction, we then segmented the log-norm contour for each word. The average log-norm value across frames within a word was taken as the word-level energy feature.

We cap the 5 dimensions based on percentiles. The percentiles of capping the five dimension are listed below (in %):

|            | Duration | Log-F0 Range | Log-F0 Median | Log-F0 Slope | Energy |
|------------|----------|--------------|---------------|--------------|--------|
| Lower cap  | 0.1      | 0            | 0.1           | 0.5          | 0.1    |
| Upper cap  | 99.9     | 99.9         | 99.9          | 99.5         | 100    |

Note that although the text transcription of the sentence is already provided in the preceding `[Text]` section, each word is repeated before the 5-dim prosody vector. This is for two reasons. First, LMs are not good at counting. If the word identity is not specified before the 5-dim prosody vector, the subsequent LM would have to align the prosody vectors to each word by itself, which is a challenging task. Second, in many cases, the written form and the speech form of an utterance are different. For example, 'In 1996' should be uttered as 'in nineteen ninety six'. This is the canonical text normalization in text-to-speech (TTS) systems. Repeating the word helps the LM to anchor the prosody information to the normalized text, rather than the unnormalized text.

Therefore, it is worth emphasizing that the text transcription in the `[Text]` section and the repeated words in the `[Prosody]` section are different. The former is in unnormalized text format with punctuations, which would make better use of the LM's inherent ability to process natural, unnormalized text, whereas the latter is in normalized text format without punctuations.

## B  Implementation Details about the Speech Decoder

The original synthesis pipeline of STYLETTS2 consists of three modules: ❶ a diffusion-based prosody style generator to generate an abstract prosody style vector, ❷ duration and prosody predictors, which produce fine-grained prosody contours conditional on the prosody style

vector, and ❸ a decoder that produces the audio based on the input text and the predicted prosody contours. As shown in Figure xx, we make two modifications. *First*, we remove the prosody style generator. *Second*, the prosody predictors are now conditional on the word-level prosody information generated by the LM.

Specifically, the duration predictor, $F_d$, and prosody predictor, $F_p$, in STYLETTS2 now become:

$$\hat{d}_{phone} = F_d(h_{phone}, d_{word}), \quad [\hat{f}_{frame}, \hat{e}_{frame}] = F_p(h_{frame}, [f_{word}, e_{word}]), \quad (2)$$

where $\hat{d}_{phone}$, $\hat{f}_{frame}$, and $\hat{e}_{frame}$ represent the phone-level predicted duration, frame-level predicted F0 and energy, respectively. $d_{word}$ $f_{word}$, and $e_{word}$ represent the word-level duration, F0, and energy information as predicted by the preceding LM. Recall that $f_{word}$ contains three dimensions: Log-F0 range, Log-F0 median, and Log-F0 slope. $h_{phone}$ and $h_{frame}$ represent the phone-level and frame-level phone embeddings, respectively, where the latter is simply the former repeated to the frame level.

The conditioning is implemented as follows. The word-level prosody tokens are passed through embedding layers, and the output embeddings are then repeated to the phone level (for the duration predictor) or the frame level (for the prosody predictor), and then directly added to the input phone embeddings. The two predictors are re-trained with the same objectives as in Style-TTS2, on a subset of Librilight. To speed up training, we initialize the weights to the original duration and prosody predictors. Note that only these two predictors are re-trained. For the other modules we adopt from STYLETTS2, we use the pre-trained version as is.

Since the word also includes silence `<Sil>`, where only the duration feature is meaningful. Therefore, we set the F0 features and energy features to a special token.

One final note is that the STYLETTS2's decoder also takes an acoustic embedding as the input, which determines the voice. In our generation process, we derive the acoustic embedding from a reference speech of our selected speakers. Note that this only affects the voice, but not the prosody, of the generated speech utterance, because neither the generation of word-level prosody (by the LM) nor the generation of the finer-grained prosody (by the prosody predictors) see the acoustic embedding, nor embeddings or any form, from the reference speech.

## C  Pre-training Details for PROSODYLM

Example paraphrases of the instruction are listed below:

- Create a story
- Spin a narrative
- Keep the narrative going
- Compose an audiobook
- Let this inspire your audiobook

## D  Experiment Details

### D.1  Configurations

All the group-A baselines were pre-trained on 29.9k hours of audiobooks from Librilight. All models were trained for 3 epochs with a batch size of 64, an initial learning rate of 1e-4, a warm-up ratio of 0.1, and a cosine learning rate scheduler. Our proposed method, ProsodyLM, as well as the baseline MIMI-tok, were trained using less than 2000 H100 GPU hours. The GLM4V-tok model was trained using less than 1000 H100 GPU hours.

For all the baselines that can control for the same speakers, we used 20 speakers, 10 male and 10 female, randomly chosen and cloned from Librilight. All experiments in Section 4.2

use the same 20 speakers. For baselines that cannot control for the same speakers, we select 10 speakers (5 male and 5 female) for ELEVENLABS, 10 speakers (5 male and 5 female) for GPT-4O, and only one speaker for GLM-4V (because it only supports one speaker).

## D.2 Baselines

• STEP-AUDIO. We follow the official repository of Step-Audio[5] and utilize the publicly available `Step-Audio-TTS-3B` checkpoint[6] for text-to-speech synthesis. The same male speakers (10 in total) and female speakers (10 in total) from Librilight, utilized for PROSODYLM evaluations, are selected to provide the speaker info/prompt for `Step-Audio-TTS-3B`. We do not change the default system prompt[7], as preliminary experiments show more mispronunciation when the default system prompt is changed (check by a Whisper-v3 ASR model, and then by a human). The text input to the model consists only of the sentences/paragraphs to be read aloud, and we do not provide any prosody/emotion labels in addition to the input text. To facilitate the analysis in Sections 4.2.2 and 4.2.3, we further segment the speech into sentences using forced alignment on the emission probability matrix generated by the `WAV2VEC2_ASR_LARGE_LV60K_960H` model from `torchaudio`.

• ELEVENLABS. For the ElevenLabs baseline, we select the `eleven_multilingual_v2` model within the ElevenLabs API. Five male and five female voices are selected for evaluation, and a distinct seed is assigned to each speaker. The output format is chosen as 16-bit, 24 kHz PCM. All other parameters of the API are kept as default. The input to the API consists only of the sentences/paragraphs to be read aloud. The output is then carefully checked for errors with a Whisper-v3 ASR model, and then by a human. To facilitate the analysis in Sections 4.2.2 and 4.2.3, we further segment the speech into sentences using forced alignment on the emission probability matrix generated by the `WAV2VEC2_ASR_LARGE_LV60K_960H` model from `torchaudio`.

• GPT-4O. For the GPT-4o baseline, we use the Openai chat-completion API with the `gpt-4o-audio-preview-2024-12-17` checkpoint. We select five male and five female preset speakers for evaluation. The temperature is set to 0.7 to encourage accurate realization of the provided transcript, while all other parameters remain at their default values. We use the following system prompt:

> Read the given paragraph. Maintain the exact text and punctuation as given without any alterations.

The model is then provided with a paragraph to read aloud. To ensure that prosody is evaluated without interference from transcription errors, we use a Whisper-v3 ASR model to assess the alignment between the generated speech and the reference text. While minor deviations are tolerated, we regenerate the audio if the ASR output reveals significant omissions or substitutions of words. While transcription accuracy is not the primary target of our evaluation, maintaining a close match to the transcript is essential to ensure that prosodic analysis is not confounded by incorrectly realized lexical content. To facilitate sentence-level analysis, we further segment the speech into sentences using forced alignment on the emission probability matrix generated by the `WAV2VEC2_ASR_LARGE_LV60K_960H` model from `torchaudio`.

## D.3 Content→Prosody

### D.3.1 Direct Style Specification

The 10 quotes used in this experiment were obtained by prompting CHATGPT, which are listed as follows

---

[5] https://github.com/stepfun-ai/Step-Audio
[6] https://huggingface.co/stepfun-ai/Step-Audio-TTS-3B
[7] https://github.com/stepfun-ai/Step-Audio/blob/main/tts.py

- "We all have our reasons for the choices we make"
- "Time moves forward, whether we're ready or not"
- "Every path leads somewhere, even if we can't see it yet"
- "People see what they want to see, nothing more"
- "Change happens, whether we embrace it or resist it"
- "A single moment can shape everything that follows"
- "Not every question has a simple answer"
- "The past is always part of the present"
- "What we do today shapes what comes next"
- "Some things are understood without being spoken"

After the speech utterances are obtained, we measure the F0, duration, and energy as follows (note that the method is different from the method to compute prosody tokens):

**Duration.** We performed forced alignment for each utterance using the `torchaudio` forced alignment pipeline with the `WAV2VEC2_ASR_LARGE_LV60K_960H` model. This provided the duration of each utterance. Given the corresponding text, we counted the number of symbols in each utterance and computed the symbol rate as the ratio between the number of symbols and the utterance duration. This metric serves as a duration-related feature at the utterance level.

**F0.** We extracted the F0 contour of each utterance using `crepe`. Using the start and end timestamps of the sentence (obtained from the forced alignment step), we segmented the F0 contour to isolate the portion corresponding to the sentence of interest. The mean F0 value over this segment was then computed and used as the utterance-level F0 metric.

**Energy.** Similar to the extraction of energy features (Appendix A), we first computed the spectrogram, followed by the log-norm of the spectrogram. We then averaged the log-norm values over the target utterance to obtain the utterance-level energy measure.

### D.3.2 Indirect Style Inference

The descriptions of six characters are generated by prompting ChatGPT. The character description of the high-voiced character is

> The little fox twitched its ears and bounced on its paws, its golden fur standing on end with excitement. Every movement was sharp and sudden, a creature wound too tight, barely able to contain its own energy. It flicked its tail with a nervous sort of eagerness, as if every thought it had was trying to escape all at once.

The character description of the low-voiced character is

> The giant's chest rose and fell like the tide, steady and unhurried, his presence casting long shadows across the firelit hall. When he turned to look at you, it was slow, deliberate, like mountains shifting under their own weight. His lips barely moved when he spoke, as if his words were carved from stone rather than formed from breath.

The character description of the fast character is

> Her hands never stopped moving, flicking through pages, tucking stray hairs behind her ears, tapping impatient rhythms on the table. Her eyes darted from face to face, searching for the next gap in conversation before one had even opened. By the time you'd processed her last sentence, she was already three ideas ahead, halfway through explaining something new.

The character description of the slow character is

The old man moved like a clock winding down, his fingers tracing invisible patterns in the air before he spoke. Deep lines carved his face, each crease a reminder of a thought considered longer than necessary. When he finally met your gaze, his eyes held the weight of unspoken words, as if time itself bent around his pauses.

The character description of the loud character is

The moment he entered the room, everything else seemed quieter by comparison, the scrape of chairs, the murmur of voices, even the distant clatter of rain against the windows. His laughter hit like rolling thunder, his gestures wide enough to command attention from across the street. When he clapped you on the back, it felt less like a greeting and more like a declaration that you were now part of the show.

The character description of the quiet character is

She walked like a shadow slipping between the cracks of the world, her presence barely more than a breath against the air. If you weren't paying attention, she would already be standing next to you, head tilted slightly as if she had been listening long before you even noticed. The corners of her lips curled in knowing amusement, as if she carried secrets meant only for those who bothered to lean in close.

### D.3.3  Clarification

The 10 simple sentences, as well as the misunderstanding is listed as follows.

- Emma locked the door reluctantly
  **Subject misunderstanding:** Oliver
  **Verb misunderstanding:** kicked
  **Object misunderstanding:** chest
  **Adv misunderstanding:** with determination
- Nathan tossed the keys impatiently
  **Subject misunderstanding:** Sophia
  **Verb misunderstanding:** shook
  **Object misunderstanding:** wallet
  **Adv misunderstanding:** with a smirk
- Isabella pushed the chair forcefully
  **Subject misunderstanding:** James
  **Verb misunderstanding:** dragged
  **Object misunderstanding:** bookshelf
  **Adv misunderstanding:** halfheartedly
- Lucas dropped the book absentmindedly
  **Subject misunderstanding:** Mia
  **Verb misunderstanding:** flipped
  **Object misunderstanding:** envelope
  **Adv misunderstanding:** with a loud thud
- Lily folded the letter carefully
  **Subject misunderstanding:** Daniel
  **Verb misunderstanding:** tore
  **Object misunderstanding:** photograph
  **Adv misunderstanding:** in frustration
- Ethan slammed the notebook shut angrily
  **Subject misunderstanding:** Grace
  **Verb misunderstanding:** flung
  **Object misunderstanding:** laptop shut
  **Adv misunderstanding:** in amusement
- Ava placed the candle on the shelf
  **Subject misunderstanding:** Henry
  **Verb misunderstanding:** knocked over
  **Object misunderstanding:** figurine
  **Adv misunderstanding:** on the windowsill
- Benjamin adjusted the tie nervously
  **Subject misunderstanding:** Chloe
  **Verb misunderstanding:** ripped
  **Object misunderstanding:** his cufflinks
  **Adv misunderstanding:** with a sigh of relief
- Zoe picked up the photograph hesitantly
  **Subject misunderstanding:** Liam
  **Verb misunderstanding:** stared at
  **Object misunderstanding:** the necklace
  **Adv misunderstanding:** with a trembling hand
- Noah wiped the glasses clean methodically
  **Subject misunderstanding:** Hannah
  **Verb misunderstanding:** polished
  **Object misunderstanding:** the mirror
  **Adv misunderstanding:** in a hurry

### D.3.4 Subjective Test

The texts for the subjective listening experiment were generated by GPT-4o model using the following prompt:

> Below are some best-selling novels. 1. Twilight by Stephenie Meyer 2. The Hunger Games by Suzanne Collins 3. HP 4. The Girl with the Dragon Tattoo by Stieg Larsson 5. The Kite Runner by Khaled Hosseini 6. The Fault in Our Stars by John Green 7. Gone Girl by Gillian Flynn 8. Where the Crawdads Sing by Delia Owens 9. The Book Thief by Markus Zusak 10. The Girl on the Train by Paula Hawkins Can you find 40 excerpts from these novels, each following the requirements below:
> 1. The excerpt should be 4-8 sentences in length
> 2. The excerpt should contain a mixture of dialogues and narrations. The dialogue should be at least one reciprocal round. The narration should be at least one complete sentence.
> 3. The excerpt should be expressive and dramatic

The subjective MOS listening test was held on AMT crowd-sourcing platform. The enrolled *master* (Sodré & Brasileiro, 2017) subjects were presented with a randomized sequence of 36 slides, featuring a single audio sample of 15-25 seconds (equally distributed over the systems under test) accompanied with its transcription text. 20 independent votes are collected per stimuli, resulting in 720 votes per system per test. The subjects were asked to select categorical answers to the following questions on a 5-grade Likert scale (*Bad, Poor, Fair, Good, Excellent*):

> **1.** Rate the overall quality and naturalness. Try to judge the quality and naturalness of the utterance rather than a specific speaker's pleasantness and personality.

> **2.** Rate the spoken intonation, based on how well it fits the context of a dialog

All experiments were conducted on AMT (Amazon Mechanical Turk) crowd-sourcing platform with votes collected from 37-39 English-native subjects.

### D.4 Prosody→Content

#### D.4.1 Emphasis Detection

We used the EmphAssess dataset, which contains 300 unique sentences spoken by 4 speakers (2 male and 2 female). In each sentence, certain words are emphasized by different speakers during reading.

To compute $q(W = w|u)$, if the target word only consists of one token, then the probability is simply the probability of outputing that token, given the context $u$. If the target word contains multiple tokens, we compute the probability by multiplication. Formally, denote $w_t$ as the $t$-th token of the work $w$, then

$$q(W = w|u) = \prod_t p(W_t = w_t|w_{1:t-1}, u). \tag{3}$$

#### D.4.2 Emotion Recognition

We used the JL corpus, an acted emotional speech dataset that includes five primary emotion categories and five secondary ones. We selected all the five primary categories (angry, neutral, sad, happy, and excited). The dataset contains recordings from four speakers (two male and two female), each reading 30 unique sentences in two sessions.

The metric we use for emotion detection is slightly different from that for emphasis detection. We do *not* aggregate across words. Instead, we compute this difference separately for each emotion word. Formally, let $\mathcal{U}$ be a set of four parallel utterances. For each emotion word, $w$, we compute

$$\mathbb{E}_{\mathcal{U}}\left[\mathbb{E}_{\{u \in \mathcal{U}: w \text{ matches emotion}\}} \log q(W = w|u) - \mathbb{E}_{\{u \in \mathcal{U}: w \text{ does not match emotion}\}} \log q(W = w|u)\right]. \tag{4}$$

|                   | Slow | Fast | High | Low | Loud | Quiet |
|-------------------|------|------|------|-----|------|-------|
| Slow Description  | 10   | 0    | 0    | 1   | 0    | 0     |
| Fast Description  | 0    | 11   | 0    | 0   | 0    | 0     |
| High Description  | 0    | 0    | 11   | 0   | 0    | 0     |
| Low Description   | 1    | 0    | 0    | 10  | 0    | 0     |
| Loud Description  | 0    | 0    | 0    | 0   | 11   | 0     |
| Quiet Description | 0    | 0    | 0    | 0   | 0    | 11    |

Table 5: Confusion matrix of human judgments (rows: intended description, columns: perceived style).

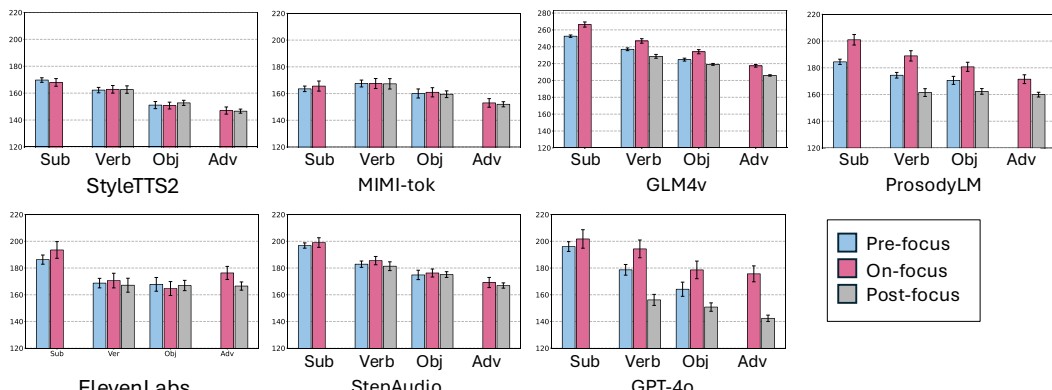

Figure 3: Average F0 under different focus conditions for all methods.

# E  Additional Experiments Results

## E.1  Indicativeness of Character Style in Indirect Style Inference

To test whether the character descriptions are truly indicative of the speaking styles, we conducted a human study. Each subject was presented with one of the character descriptions and asked to choose one from six speaking styles: *high-voiced*, *low-voiced*, *fast*, *slow*, *loud*, or *quiet*.

We invited 11 AI researchers (who are not authors and are unfamiliar with the project) to participate in this test. The confusion matrix is shown in Table 5, where each row corresponds to a description, and each column represents the perceived speaking style.

As shown in Table 5, subjects were able to choose the correct speaking styles with almost 100% accuracy, except that one subject confused *slow* with *low*. This result verifies that the character descriptions are strongly indicative of speaking styles as perceived by human readers.

## E.2  Clarification

**Contrastive Focus.**   Figure 2 shows the average F0 of different sentence components for different focus conditions. Due to space limitations, we displays only a subset of the methods. The full results is shown in Figure 3.

**Speech Rate Adjustment.**   Another typical human behavior during clarifications is to slow down. To test whether PROSODYLM also captures this behavior, we compute the average symbol rate in the last quote versus the first quote. The results, shown in Figure 4, verify that PROSODYLM indeed slow down, with the symbol rate reduced by 5%. On the other hand, it is quite surprising to observe that most of the baselines would speed up during

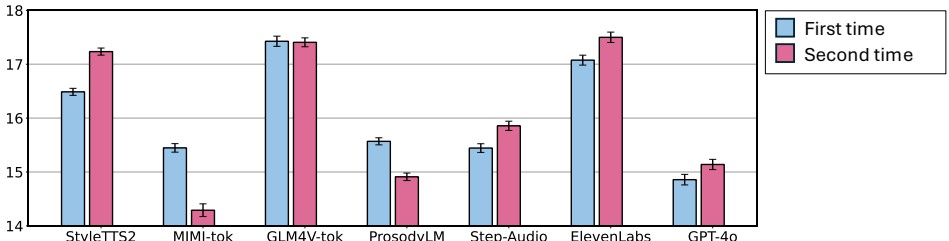

Figure 4: Symbol rate comparison between first-time and second-time uttering the sentence during the classification setting.

| Methods | Direct Style Specification | | | Indirect Style Inference | | |
|---|---|---|---|---|---|---|
| | F0 pair | Dur. pair | Energy pair | F0 pair | Dur. pair | Energy pair |
| w/ [Global] | 18.5 (1.79) | 1.48 (0.17) | **0.171** (0.058) | 21.88 (1.15) | 1.17 (0.14) | **0.061** (0.016) |
| w/o [Global] | **23.9** (2.45) | **1.87** (0.18) | 0.032 (0.057) | **23.36** (1.84) | **1.42** (0.12) | 0.042 (0.013) |

Table 6: Results of direct (Section 4.2.1) and indirect style following (Section 4.2.2). Numbers in parentheses are standard deviations.

the classification, including all the group-B baselines. The only baseline that learns to slow down is MIMI-tok. This indicates that learning to slow down may be harder to learn.

### E.3 Global Token

To study the effect of adding/removing the [Global] token, we repeat some experiments comparing PROSODYLM with and without the [Global] token. These experiments are the same as Tables 1, 3 and 4, listed in Tables 6, 7 and 8 respectively.

As shown in Table 6, PROSODYLM with and without [Global] perform on par — the former performs better in capturing energy and the latter in F0 and duration. As shown in Table 7, PROSODYLM with [Global] has an obvious failure in recognizing angry emotions. Similar failure is also observed in Table 8. To sum up, PROSODYLM without [Global] excels in more tasks and metrics, but PROSODYLM with [Global] has the advantage in allowing for external high-level control.

### E.4 Content→Content

#### E.4.1 Perplexity

We evaluate the text-section perplexity on a held-out set of validation audiobooks (unseen during continuous pre-training), and compare it with two text-only LLM baselines: (1) Llama3.1-8b-Instruct, and (2) Llama3.1-8b-Instruct continuously fine-tuned on the same set of audiobooks. The results are summarized in Table 9.

As can be observed, ProsodyLM only increases perplexity by 2 compared to the text-only LLM fine-tuned on the same data, which indicates that introducing prosody tokens has only minor effects on text modeling capabilities.

#### E.4.2 Intent Classification

We perform the intent classification (IC) experiment to further evaluate the text processing capability of PROSODYLM. We evaluate our method on the test set of the fluent_speech_commands_dataset[8]. Recall that since ProsodyLM is not instruction-tuned, we need to modify the task as a next-token prediction task (as in Sections 4.3.1 and 4.3.2).

---

[8] https://www.kaggle.com/datasets/tommyngx/fluent-speech-corpus

| Methods | Emphasis | Emotion Recognition | | | |
|---|---|---|---|---|---|
| w/ [Global] | 0.0494 (0.0094) | 0.048 (0.008) | **0.311** (0.016) | 0.114 (0.013) | -0.036 (0.008) |
| w/o [Global] | **0.0656** (0.0062) | **0.065** (0.008) | 0.273 (0.016) | **0.177** (0.017) | **0.020** (0.009) |

Table 7: Average increase in log output probability in emphasis detection (Section 4.3.1) and emotion recognition (Section 4.3.2). Numbers in parentheses are standard deviations.

| Methods | F0 Pair | Dur. Pair | Energy Pair |
|---|---|---|---|
| w/ [Global] | 34.5 (2.9) | 0.46 (0.14) | -0.004 (0.064) |
| w/o [Global] | **47.9** (3.6) | **1.09** (0.14) | **0.038** (0.047) |

Table 8: Dialogue continuation results. Numbers in parentheses are standard deviations.

Specifically, we adopt the following template for the three aspects of IC—*action*, *object*, and *location*, respectively:

```
'Turn off the lights.' [Prosody] Based on what the user said, the intention is ___
'Turn off the lights.' [Prosody] Based on what the user said, the object is ___
'Turn off the lights.' [Prosody] Based on what the user said, the location is ___
```

We then probe the ProsodyLM's output probabilities for a list of candidate continuations and determine whether the correct answer has the highest probability.

We include two baselines for comparison: `Llama3.1-8b-Instruct` and `Llama3.1-8b-Instruct` continuously fine-tuned on the same set of audiobooks as ProsodyLM. The second baseline is nearly identical to ProsodyLM, except that the `[Prosody]` section is not present during fine-tuning. This allows us to directly assess the effect of the `[Prosody]` section on performance in this task.

The results are shown in Table 10. As can be observed, ProsodyLM does not significantly underperform the text-only LLMs, implying that introducing the `[Prosody]` section does not substantially harm performance, even in a task like intent classification.

Note that these results are not directly comparable to prior work in the literature, as our evaluation is adapted to a next-token prediction setting suited for audiobook completion.

| Model | Llama3.1-8b | Fine-Tuned | ProsodyLM |
|---|---|---|---|
| Perplexity | 24.53 | 11.70 | 13.87 |

Table 9: Perplexity on validation audiobooks.

| Aspect | Llama3.1-8b-Instruct | Fine-Tuned | ProsodyLM |
|---|---|---|---|
| Action | 91.15 | 87.02 | 87.88 |
| Object | 87.35 | 86.26 | 93.62 |
| Location | 94.81 | 95.02 | 87.91 |
| Average | 91.10 | 89.43 | 89.80 |

Table 10: Intent classification accuracy (%) on Fluent Speech Commands.

