# OpenReview forum: "ProsodyLM: Uncovering the Emerging Prosody Processing Capabilities in Speech Language Models"
_colmweb.org/COLM/2025/Conference — COLM 2025_

### Official Review · Reviewer_GTN9 · 2025-05-09

**Rating:** 6
**Confidence:** 4
**Ethics Flag:** 1

**Summary:**

This paper proposes ProsodyLM, a speech language model that explicitly tokenizes the prosody information in addition to standard text processing. The prosody tokens are based on the StyleTTS2 system's prosody features, in order to facilitate decoding to expressive speech. Specifically, the prosody token sequence is appended to the text token sequence (in the backbone LM) and includes: pause duration (between words), duration of phones, 3 log-F0 contour features, and log-energy.

ProsodyLM is evaluated on several tasks to get at its ability to learn dependencies between prosody and content, e.g. whether a word is correctly emphasized to signal contrastive focus, emotion consistency, and speaking style consistency. Baselines for comparison include 3 open-source models with similar amount of training data, and 3 (2 of which are commercial) systems trained on larger amounts of data. The evaluations show competitive performance by ProsodyLM even with commercial systems.

**Questions To Authors:**

- Lines 65-67: "Second, most speech tokens **may be too abstract** to the LLM, and it may take an LLM a significant portion of its representation power just to re-align the speech tokens." --> could you clarify what you mean here?
- Lines 319-321: "Only two group-A baselines are included here because it is **infeasible to fit the rest into the Prosody-->Text generation** mode." --> what does this mean?

**Reasons To Accept:**

- The idea of incorporating prosody tokens into a typical LM framework is well motivated and this seems to work in this case, especially considering the smaller amount of data used in pretraining here compared to commercial models.
- The system is built on several pretrained open source and successful components (StyleTTS2, LLaMA3.1-8b-Instruct), so it should be reproducible and valuable to the research community.
- The paper is reasonably clear and easy to follow (some exceptions listed below). The demo page also clarifies the input processing step better and the audio samples support the authors' claims, at least based on my listening and subjective evaluation.

**Reasons To Reject:**

My major concerns and questions are regarding certain modeling choices and evaluation tasks:

1. Quantization of prosody tokens:
- In lines 156-157, it is described that the (originally continuous) features such as duration, log-F0 contour features, and energy are normalized and quantized into 512 bins, which to me seems like introducing extra steps. I understand that "tokenization" involves discretization, but in practice, when the features are used in the model, the tokens themselves are embedded into continuous vectors again, so why not use some sort stacking/pooling for these frame-based features (+necessary projections) directly in the inner network layers instead of doing the continuous->discrete->continuous transformation?
- Duration, F0, and energy features can vary quite a lot between phones within a word, so doing word-level averaging seems problematic, especially when this averaging is done at the feature processing step, i.e. what ProsodyLM receives as input is already the coarser version of these features, instead of e.g. having another learnable network to learn these time-varying characteristics. As a simple example, vowels have higher energy and duration than consonants so doing averaging over the whole word (which might have more consonants than vowels) would affect these statistics. Similarly, the F0 slope modeling as a straight line also seems problematic. It's typical to have a concave/convex-looking F0 contour (if not more complex) so linear modeling might discard important information.
2. Evaluation tasks:
Most of the tasks seem new (to me) and not typically used in related work. This is not necessarily a reason for criticizing the paper, especially when the authors are trying to get at prosodic phenomena not looked at in Speech LM and TTS work. However, if the task itself is new, there should be sufficient motivation and grounding in literature to support it. In the current manuscript, the way some evaluation tasks are presented seem a bit arbitrary and not consistent with what's introduced and motivated in the Introduction.
- Indirect style inference: the descriptions of characters were generated by GPT-4o, have these been verified to be indicative of speaking styles inferable by an average person?
- Clarification: the authors give results of F0, duration, energy computed for pre-, on, and post-focus words; these are aggregated across utterances in the (evaluation) data and compared with other speech generation models. I maybe misunderstanding, but in this setting, it is entirely possible for one model to generate higher F0/duration/energy _on average_ so the inter-model comparison might not be appropriate. Do you have these results in terms of comparative difference with other words in the same sentence, i.e. whether the feature is statistically significantly (and perceptually) different for on focus words vs. others?
- Emphasis detection and emotion recognition: if I understand correctly, this task can be evaluated like a multi-class classification and scores such as accuracy and F1 (for the correctly chosen emotion category) can be reported. It's not clear to me why the authors chose to use difference in log-prob instead.
- Prosody to prosody: this task is the least convincing to me. The evaluation was essentially based on whether F0/energy/duration average _difference_ is preserved from the "prompt" pair to the output pair, which seems could be learned in the model just based on inherent speaker encoded information. This task is also least consistent with the original motivation (e.g. in lines 48-49), where the goal was to be able to adopt appropriate tone as a conversation partner and cannot be evaluated in this configuration.

Overall, my main concern for these evaluation schemes is that I fail to see the connection with (a) other work grounded in linguistics/prosody in human communication, and (b) consistency with the authors' claimed motivations. I would also be interested in seeing human evaluation of these tasks (which is labor intensive and costly, so this is just an added comment) - e.g. given a generated utterance, whether listeners indeed perceived a certain emotion or if it really served the clarification purpose intended in Section 4.2.3

Minor nits and other typographical comments:
- Lines 270-271: should Xu (1999) really be the citation for contrastive focus? I believe this is a very old area of study in linguistics, at least dating back to Chafe (1976) ("Givenness, Contrastiveness, Definiteness, Subjects, Topics and Points of View"). Of course the "correct" cite is not necessarily the oldest one, but there's a reason the canonical citation for LSTM is Hochreiter and Schmidhuber (1997) instead of any paper from 2010s.
- Line 117: typo "Speech Encoer" --> "Speech Encoder"
- Table 1's caption is split to the top of both pages 6 and 7, for page 7 it now appears under Figure 2. Table 1 also looks to be over the margin (right)
- Line 112: Mini-Omni citation is wrong; Jawaid et al. citation is duplicate

---

> ### Author Response · Authors · 2025-06-02
> **Response to Reviewer GTN9 (Part 1)**
>
> Thank you for your detailed and constructive feedback! We would like to address your concern first and then answer your questions.
>
> Regarding your concerns -
>
> ### 1.1 Why do continuous -> discrete -> continuous conversions?
>
> The process of first discretizing the prosody features and re-projecting them into continuous space may seem like a redundant process, but it is actually a key step to learning **multi-modal distributions** (i.e., probability distributions with multiple peaks) for prosody. More specifically, for the same content, there could be multiple prosody styles to utter the content. As a result, the distribution of prosody features tends to have multiple modes (or multiple peaks). Turning the prosody features into discrete bins, together with the cross-entropy loss, allows the ProsodyLM to directly learn the PMF (probability mass function) itself, which can accommodate aan rbitrary number of modes.
>
> On the other hand, if we directly ask the LLM to predict the continuous prosody features, using simple losses such as $l_2$ or $l_1$ loss, it would lead to **mode collapse**, i.e., lumping all the peaks into one, because $l_2$ or $l_1$ are essentially fitting a Gaussian or Laplacian distributions, respectively, both of which are uni-modal distributions. The benefit of discretization on learning multiple modes is also documented in [R1].
>
> An alternative way to learn multi-modal distributions without discretization is to fit mixture models, such as Gaussian mixture model or Mixture of Logistics, using the maximum likelihood objective [R2]. However, we decided to use the discretization approach because it unifies the text learning losses and prosody learning loss, which enables a stable convergence behavior.
>
> We are happy to add our above discussion to the paper.
>
> [R1] Van Den Oord, Aaron, et al. "Wavenet: A generative model for raw audio." _arXiv preprint arXiv:1609.03499_ 12 (2016).
> [R2] Oord, Aaron, et al. "Parallel wavenet: Fast high-fidelity speech synthesis." _International conference on machine learning_. PMLR, 2018.
>
> ### 1.2 Word-level simple averaging is sub-optimal
>
> Thank you for the suggestion! Introducing another network to learn the time-varying prosody feature could improve the prosody modeling, and we would love to pursue this as a future direction.
>
> Meanwhile, we just wanted to comment that simple averaging is not that bad in many cases. This is because the decoder (Section 3.2), which is responsible for recovering the frame-level prosody contours from the word-level ones, has sufficient capabilities and information access to recover the potential information loss of simple averaging. Specifically, the F0 & energy predictors both take the word-level prosody feature AND the full phoenemized script as their inputs (lines 186-188), so it has the capability to account for 1) the influence of vowel/consonants composition on the average energy, and 2) more sophisticated F0 contours due to syllable stress, phone identity, etc.
>
> To further verify this, in [F0 reconstruction plot](https://raw.githubusercontent.com/anonymous0818/anonymous0818.github.io/refs/heads/main/rebuttal/f0_recon_plot.png) and [energy reconstruction plot](https://raw.githubusercontent.com/anonymous0818/anonymous0818.github.io/refs/heads/main/rebuttal/energy_recon_plot.png), we plot example prosody contours of **an unseen test utterance** reconstructed from word-level simple averages by our learned decoder (green line), against the ground-truth prosody contours (blue line). We further plot the word-level average information as a red dashed line. As can be observed, even though the word-level average loses much information, it still retains the key information that contributes to a decent reconstruction quality. Notably, in the F0 reconstruction plot, the ground-truth F0 for 'gentleman' is convex, and the decoder and successfully reconstructs the convexity even though the word-level average loses the information. In the energy reconstruction plot, there is a significant unvoiced portion in the word 'gentlement', dragging down the word-level average, but the reconstruction result manages to recover the voiced energy.

---

> > ### Author Response · Authors · 2025-06-02
> > **Response to Reviewer GTN9 (Part 2)**
> >
> > ### 2.1 Human verification of character descriptions in 'Indirect Style Inference'.
> >
> > To test whether the character descriptions are truly indicative of the speaking styles, we add a human study where we present each one of the character descriptions to each subject and ask them to choose one from six speaking styles: high-voiced, low-voiced, fast, slow, loud, quiet. We invited 11 AI researchers (who are not authors nor familiar with this project) for the test. The confusion matrix is listed below, where each row corresponds to a description, and each column represents a perceived style.
> >
> >
> > |	| Slow	| Fast	| High	| Low	| Loud	| Quiet	|
> > |-----|-----|-----|-----|-----|-----|-----|
> > | **Slow Description** | 10 | 0 | 0 | 1 | 0 | 0 |
> > | **Fast Description** | 0 | 11 | 0 | 0 | 0 | 0 |
> > | **High Description** | 0 | 0 | 11 | 0 | 0 | 0 |
> > | **Low Description** | 1 | 0 | 0 | 10 | 0 | 0 |
> > | **Loud Description** | 0 | 0 | 0 | 0 | 11 | 0 |
> > | **Quiet Description** | 0 | 0 | 0 | 0 | 0 | 11 |
> >
> > As can be observed, subjects were able to choose the matching styles with almost 100% accuracy, except that one subject was confused between 'slow' and 'low'. This verifies that the descriptions are indicative of speaking styles perceived by human readers.
> >
> > Meanwhile, Appendix D.3.2 lists all the character descriptions. We welcome readers to judge their alignment with the intended style.
> >
> > ### 2.2 Inter-model comparison in the 'contrastive focus' experiment.
> >
> > Thank you for pointing this out, and you are absolutely right that different models may produce different ranges of prosody contours.
> >
> > That is why we did not directly compare the absolute F0/energy contours among different models, but rather the **relative prosody difference** among on-focus, pre-focus, and post-focus words **within the same model**. More specifically, in Figure 2, what we are comparing are the relative height differences in the blue, red, and grey bars within the same group. As can be observed, only ProsodyLM and GPT-4o have significant height differences, and these differences are all greater than the confidence intervals marked by the black lines, which indicates statistical significance. The other models have little difference. On the other hand, different models do have different height ranges, but that is not our focus.
> >
> > ### 2.3 Metrics in emphasis detection and emotion recognition
> >
> > Thank you for bringing this up! As also noted by Reviewer HdRD, one major complication of both emphasis detection and emotion recognition tasks is that the answers depend not only on the prosody of the quote, but also the **content** of the quote. Since the datasets are designed to be parallel (same content paired with different emotions/emphasis words), the prosody cues and content cues are often in conflict. Consider the following example (question 3 from Reviewer HdRD)
> >
> > ```
> > 'And that to me was so exhilarating.' He/She emphasized ‘___’
> > ```
> > The content of the quote would strongly suggest 'exhilearting' should be emphasized, which creates a significant bias towards certain words even if these words were not prosodically emphasized.
> >
> > As a result, we need to introduce metrics other than simple F1 scores to remove such biases. That is why we applied the **relative log-probability differences** as our metrics. Specifically, take emphasis detection as an example, instead of measuring whether ProsodyLM outputs the emphasized word with the highest probability, we measure the probability difference of outputting a word when the word is emphasized v.s. when it is not. This would separate out the effect brought by prosody changes, controlling for the content influence (because the content is the same when calculating the relative probability difference). We did not include this rationale due to space limitations, but we will add it back to the paper.

---

> > ### Author Response · Authors · 2025-06-02
> > **Response to Reviewer GTN9 (Part 3)**
> >
> > ### 2.4 Task selection rationales and consistency with motivation
> >
> > Finally, we would like to comment on your general concerns about our evaluation task selection.
> >
> > **First**, although we did not elaborate on this in the paper due to space limitations, we do have clear rationales behind our evaluation tasks. Specifically, there are three considerations for our task design:
> >
> > 1. We identified two important aspects of prosody, **emotion** and **emphasis**, which both serve key communicative purposes in human speech interactions, and which are covered in both speech analysis and generation benchmarks.
> > 2. These two aspects, paired with the 'content->prosody', 'prosody->content', and 'prosody->prosody' settings, can be projected to the following canonical speech processing tasks.
> >
> > |  | Content -> Prosody | Prosody -> Content | Prosody -> Prosody |
> > |----------|----------|----------|----------|
> > | **Emotion** | Expressive Speech Synthesis  | Emotion Recognition  | Style Cloning |
> > | **Emphasis**  | Emphasis Synthesis  | Emphasis Detection  | - |
> >
> > 3. We then need to adapt the tasks to audiobook settings that involve sophisticated semantics understanding and inform practical scenarios. Hence
> > -- ***Expressive speech synthesis*** is adapted by blending the style specification in an audiobook context, hence deriving ***direct style specification*** and ***indirect style inference***.
> > -- ***Emphasis synthesis*** is adapted by embedding the direction of what to emphasize into a ***contrastive focus*** setting.
> > -- ***Emotion recognition*** and ***emphasis detection*** are adapted into an audiobook completion task.
> > -- ***Style cloning*** is adapted into an audiobook conversation between two characters with opposite styles, with reference styles provided as the previous round of conversation.
> >
> > **Second**, the prosody->prosody task in the paper actually involves non-trivial semantic understanding capabilities, because it requires the model to switch among different styles based on the character identity. A common mistake of existing methods is that they tend to maintain the same style across different speakers, ignoring the character change. Other speech language models would sometimes mismatch the character styles (i.e., the style of Tom in the next paragraph is different from that in the previous one). This is our motivation for introducing the dialogue setting, instead of the monologue one.
> >
> > **Third**, to improve the consistency between the motivation and the actual experiments, we plan to make two changes:
> >
> > 1. Add more motivating examples to the introduction. We will add a rising tone vs falling tone example to 'Prosody->Content' (as requested by Reviewer HdRD, question 9), and the aforementioned character consistency challenges to 'Prosody->Prosody'.
> > 2. Add a new 'Prosody->Prosody' experiment closer to the intro motivation. Specifically, we use the emotion classification dataset (the JL corpus). For each utterance-emotion combination, we construct the following template
> > ```
> > “The lord swims in the sea.” [Prosody (Ref Speech)] “I know.” Tom said. [___]
> > “The lord swims in the sea.” [Prosody (Ref Speech)] “I am sorry.” Tom said. [___]
> > ```
> > We let ProsodyLM complete the prosody of the second quote. Our interesting observation is that, if the response quote is 'I know', its prosody varies greatly when the emotion of the reference quote changes. If the reference quote is ***angry***, the reponse quote would sound ***angry and arrogant*** ([example audio 1](https://www.dropbox.com/scl/fi/k6oab1vp8ziv3iu0343fp/female_angry.wav?rlkey=85c2gdlofl9fz77vfh66cdm8o&e=2&st=eworr7jl&raw=1) and [example audio 2](https://www.dropbox.com/scl/fi/fw46yxogvc8b22837fwho/male_angry.wav?rlkey=f3f3mz7w65qnzgdveywdzbcsm&st=358wlrxs&raw=1) of 'I know' in response to angry); if the reference quote is ***sad***, the response quote would sound ***soft*** ([example audio 1](https://www.dropbox.com/scl/fi/re2otphvqt3hsbkrzygk2/female_sad.wav?rlkey=q4mxz7oxzi01e1fwsyvvkcs6u&st=ka2uf3tn&raw=1) and [example audio 2](https://www.dropbox.com/scl/fi/kiluolde89hq3skxb9d35/male_sad.wav?rlkey=o7jyei3ew9mc904p397001f44&st=re0xhjnt&raw=1) of 'I know' in response to sad).
> >
> > On the other hand, if the response quote is 'I am sorry', it would sound ***soft*** both when the reference quote is ***angry***  ([example audio 1](https://www.dropbox.com/scl/fi/26bq854zrjcpwej4e1krz/female_angry.wav?rlkey=1c8u764yl4fpzz0o0ebw0bu6i&st=nzp3exwc&raw=1) and [example audio 2](https://www.dropbox.com/scl/fi/odirqwv20r1f6qqav756c/male_angry.wav?rlkey=te1d1ctxoqk0w8bnn49xk8gys&st=rhmu9ohd&raw=1) of 'I am sorry' in response to angry) and when the reference quote is ***sad*** ([example audio 1](https://www.dropbox.com/scl/fi/a1keaxyooaczo39gmsarr/female_sad.wav?rlkey=1f7ww4qzp7ymvc3p119h48df5&st=pxe1soiw&raw=1) and [example audio 2](https://www.dropbox.com/scl/fi/8wtkr1s26nxahsbxj7uaz/male_sad.wav?rlkey=osifk93htpna1tex3mtcerq6q&st=rzmrzdhn&raw=1) of 'I am sorry' in response to sad).

---

> > > ### Author Response · Authors · 2025-06-02
> > > **Response to Reviewer GTN9 (Part 4)**
> > >
> > > Regarding your questions -
> > >
> > > ### Q1. What do Lines 65-67 mean?
> > >
> > > By 'abstract', we meant that it is difficult to assign semantic interpretations to conventional speech tokens. On the other hand, prosody tokens have clear semantic interpretations. For example, the F0 token at larger bins can be interpreted as 'high pitch'; the energy token at larger bins can be interpreted as 'loud'. Furthermore, prosody tokens are tied to each semantically meaningful word. Since most speech language models are continuously trained from text-only LLMs. The connection of the speech/prosody tokens to the semantic space would make it easier for the LLM to acquire the speech/prosody tokens.
> > >
> > > ### Q2. What do Lines 319-321 mean?
> > >
> > > For the 'Prosody->Content' experiments, we need to feed the tokenized reference speech to the speech language model, and ask it to complete the following sentence as an audiobook. Other than the baselines we included in Table 3, the other baselines cannot accomplish the task:
> > >
> > > * StyleTTS2 and ElevenLabs are pure text-to-speech systems, which do not take audio as input.
> > > * GPT-4o and Step-Audio can take audio as input, but they are instruction-tuned to treat input audio as queries, and would provide answers to the content of the query rather than perform continuation based on the prosody. In addition, GPT-4o is a closed-source model, which makes it difficult to elicit the output probabilities other than the top-k ones.

---

> ### Author Response · Authors · 2025-06-07
> **Follow up on rebuttal discussion**
>
> Dear Reviewer GTN9, since it is approaching the deadline of the discussion period, but no discussion has been initiated so far, we would like to follow up on whether your concerns can be adequately addressed. We have provided in-depth justifications of our algorithm design and evaluation tasks. We also made changes to the writing and conducted new experiments according to your suggestions. We are happy to engage in discussion with you and answer any further questions you might have. Thank you for your time!

---

> > ### Comment · Reviewer_GTN9 · 2025-06-08
> >
> > I have read the authors' responses and other reviewers and discussions, the authors' clarifications make sense to me. I have raised my score accordingly.

---

### Official Review · Reviewer_HdRD · 2025-05-13

**Rating:** 7
**Confidence:** 4
**Ethics Flag:** 1

**Summary:**

The article describes ProsodyLM, a speech language model aimed at better prosodic modeling. Prosody modelling is important as model performance in both speech understanding and synthesis depends on its ability to capture the relations between content/semantics and prosody features.

The main contribution is the proposed token encoding method for speech input, with explicit discrete prosody tokens based on manually-defined prosodic features (word-level 5-dimension prosody values + a global token). Each speech input utterance is encoded into a sequence of text tokens (obtained with Whisper) followed by the word-level prosody tokens, and is fed to an LLM initialized with Llama3.1-8b-Instruct weights. The model is trained with the usual next-word prediction task. Experiments are performed to evaluate model capability of capturing prosody/content dependencies.

The proposed token encoding, with explicit discrete prosody tokens based on manually-defined prosodic features, seems novel. IMO, the experimental part of the work and also some parts of the method description need to be improved.

**Questions To Authors:**

1/ Section 4.2.1 and 4.2.2: I am afraid I could not understand the details of the evaluation method here. For the content->prosody dependency type, the model is supposed to take as input a text and then generate speech with the expected prosodic characteristics (i.e. TTS mode). To evaluate this capability, the 60 utterances of the test data are read by 20 speakers. Then :

> "For each F0 pair with the same quote and speaker, we calculate the difference in average 245 F0 between the two utterances in this pair (high voice minus low voice)"

Is it the difference between the 2 speech utterances generated by the model for each F0 pair (high vs low)? Or is it the difference between model-generated speech utterance and the (human) speakers' utterance? In the first case, i.e. if the speaker-produced speech utterances are not used in the evaluation, what is their purpose?

2/ Each test sample contains the quoted sentence (e.g. "Time moves forward, whether we’re ready or not!") and the explicit/implicit style specification (e.g. 'He said in a high voice.'). A good understanding of the content->prosody dependency type would allow the model to produce the two parts  with different prosodic characteristics. However, listening to some of the demo examples on https://anonymous0818.github.io/, it appears that, at least to some extent, the model generates similar prosodic features for both the quoted sentence and the citation prefix, which sounds quite unnatural. This is not captured by the evaluation method, which, to my understanding, aggregates the prosodic feature values across all the words of the whole test sample. To solve this issue, I would suggest to compute the evaluation metrics separately for the two utterance parts (the quoted sentence vs. the rest) using the speakers' speech prosody as reference.

3/ A potential issue with the prosody->content experiments, in particular in the emphasis detection task, is that since text tokens are provided in the context (including in the `[Prosody (Ref Speech)]` part), this can be a confounding factor: for instance, in the provided example ("And that to me was so exhilarating.") and without any prosody information, it is very likely that emphasis will be on "exhilarating", while it's unlikely to be on grammatical words like "to", "and" or "so"). To control for that confounding factor, I would suggest to include a pure text baseline in the experiment, leveraging an only-text LLM, e.g. the Llama3.1-8b-Instruct that was used to initialize ProsodyLM.

4/ Section 4.2.4,  What is the meaning of "Consistent Speakers" and "Inconsistent Speakers" categories in Table 2? And why are the methods evaluated in both cases not the same? I could not find any explanation of these categories in the body of the article.

5/ Section 4.3.2 (Emotion Recognition) The JL Corpus for emotion recognition was used for evaluation, however, only a subset was considered, excluding the neutral class examples (lines 704-711). Why? I would suggest to redo the experiment with the full dataset and update Table 3 accordingly.

6/ There is a risk that the proposed text+prosody token encoding would hinder model performance on other, semantically-rich tasks, like intent classification and slot filling. I would suggest to include an additional experiment for at least intent classification on speech input, with a pipelined baseline consisting of Whisper+Llama3.1-8b-Instruct in zero-shot mode. These additional experiments should be easy to run, since they do not require any training.

7/ Appendix C, lines 588-593: it is stated that a speaker-dependent information (average log-F0 of the speaker) is included in the prompt during the continuous pre-training of ProsodyLM. Was this information also used at inference time, in the experiments? What is the impact on performance for speakers unknown to the model and for whom this information is unavailable?

8/ For the pretraining on Librilight data, are the ground-truth transcriptions used (i.e. supervised pretraining)? Or do you instead use Whisper transcriptions?

9/ When presenting the 3 types of prosody/content dependencies, the example for prosody->content dependency ("asking if there is anything wrong in response to an upset tone from the user") requires other abilities than prosody understanding, namely some kind of learned simulation of empathy. I wonder why there is no mention of a simpler and more obvious case of prosody->content dependency: factual questions expressed without interrogative (grammatical) form, just with rising intonation ("It's currently raining in Miami"↑), which the model should understand as questions and not declarative statements.

10/ Appendix D4.1: Lines 704-711 describes the data used for emotion recognition and therefore should be moved to Appendix D4.2

**Reasons To Accept:**

- The proposed token encoding, with explicit discrete prosody tokens based on manually-defined prosodic features, seems novel to the best of my knowledge.

- The work also highlights the importance of the content-prosody dependencies in speech understanding and synthesis.

**Reasons To Reject:**

- Some methodological issues in the experiments, which complicates the interpretation of the results (see my questions/suggestion to the authors).

- Lack of clarity in the description of some aspects of the work  (see my questions/suggestion to the authors), and some important details are absent from the main body of the article (e.g. inclusion of speaker-dependent information in the input instruction at least during training; the exclusion of part of the data in the experiment on emotion recognition; the initialization of model weights with Llama3.1-8b-Instruct weights before the pretraining).

---

> ### Author Response · Authors · 2025-06-02
> **Response to Reviewer HdRD**
>
> We thank you for your detailed comments and questions! We will address your 'Questions to Authors' first and then further comment on your 'Reasons to Reject' section.
>
> Regarding your questions -
>
> ### 1. What role do ‘utterances read by human speakers’ play in Sections 4.2.1 and 4.2.2?
>
> We apologize for the confusion here. There are no utterances generated by humans involved in Sections 4.2.1 and 4.2.2. By **'.. each read by 20 speakers'** we actually meant **each utterance was generated in 20 different speaker voices**. The prosody differences were calculated for each speaker's voice and then averaged. The reason why we need to generate utterances in different speaker voices is that different speaker voices have different F0 levels (males have lower and females have higher). We want to verify that ProsodyLM's emergent capabilities could work across different F0 levels. To avoid the confusion, we will correct line 243 as 'each generated in 20 different speaker voices (so ProsodyLM needs to generate the prosody predictions for 20 different F0 levels)', and add the above rationales.
>
> ### 2. Prosody difference of the quoted sentence v.s. of the style specification
>
> Thank you for noting this! We also noticed that in the direct style specification experiment (Section 4.2.1), the style specification portion may undesirably carry the prosody characteristics in the quote. Therefore, we follow your suggestion and separately calculate the prosody difference of the quoted sentence (this was what's currently in Table 1), and the prosody difference of the direct style specification. The results are as follows -
>
> | Method | Sentence Part | F0 Pair | Dur Pair | Energy Pair |
> |---|---|---|---|---|
> | StyleTTS2 | Quote | 3.25 (1.77) | 0.28 (0.16) | 0.125 (0.092) |
> | | Style Spec | 3.44 (1.73) | 2.33 (0.15) | 0.841 (0.078) |
> | MIMI-tok | Quote | 1.86 (1.67) | 0.42 (0.21) | 0.126 (0.091) |
> | | Style Spec | 2.68 (2.01) | 2.97 (0.22) | 1.137 (0.098) |
> | GLM4V-tok | Quote | 5.53 (1.85) | 0.54 (0.26) | 0.037 (0.084) |
> | | Style Spec | 4.64 (1.11) | 3.26 (0.16) | 1.072 (0.062) |
> | ProsodyLM | Quote | 18.51 (1.79) | 1.48 (0.17) | 0.171 (0.058) |
> | | Style Spec | 2.21 (0.82) | 3.62 (0.18) | 0.931 (0.041) |
> | Step-Audio | Quote | 3.18 (1.52) | 1.19 (0.17) | 0.258 (0.075) |
> | | Style Spec | -1.00 (1.94) | 3.66 (0.14) | 0.910 (0.055) |
> | ElevenLabs | Quote | 7.04 (2.54) | 0.11 (0.18) | 0.165 (0.111) |
> | | Style Spec | 3.36 (3.43) | 2.82 (0.12) | 0.688 (0.038) |
> | GPT-4o | Quote | 115.36 (5.77) | 4.36 (0.28) | 1.143 (0.141) |
> | | Style Spec | 14.08 (3.04) | 5.01 (0.22) | 1.348 (0.098) |
>
> Rows labelled 'quote' show the prosody differences of the character quotes (which is what Table 1 shows); rows labelled 'style spec' show the differences of the style specification portion. We have some interesting observations. First, ProsodyLM, as well as competent baselines such as GPT-4o, has much smaller F0 differences in the style specification part, which indicates that it can clearly distinguish the quote and the style specification in terms of F0. On the other hand, however, all methods have higher duration and energy differences in the style specification part. We are interested in this and will investigate whether such a phenomenon is due to the modeling limitations or a natural phenomenon that also exists in human speech.
>
> ### 3. In 'Prosody -> Content' experiments, the generated content may also be affected by the historical content.
>
> Thank you for also noting this! We found out that for both emotion classification and emphasis detection tasks, the generated labels are simultaneously influenced by both historical prosody and the content of the quote. That is why we applied the **relative log-probability differences** as our metrics instead of absolute probability. Specifically, take emphasis detection as an example, instead of measuring whether ProsodyLM outputs the emphasized word with the highest probability, we measure the probability difference of outputting a word when the word is emphasized v.s. when it is not. This would separate out the effect brought by prosody changes, controlling for the content influence (because the content is the same when calculating the relative probability difference). Under this metric (which already addresses the confounding issue), the text-based LLM result is not applicable because it does not depend on prosod,y so the relative log-probability difference would be 0. We did not include this rationale due to space limitations, but we will add it back to the paper.

---

> > ### Author Response · Authors · 2025-06-02
> > **Response to Reviewer HdRD (Part 2)**
> >
> > ### 4. Consistent Speakers vs. Inconsistence Speakers
> >
> > This is possibly related to the confusion in question 1. Recall that our utterances were generated in different speaker voices, so to rule out the interference of speaker voices, it is most desirable if the utterances generated by the baselines are also in the same set of speaker voices. Unfortunately, not all baselines support generating in our specified speaker voices -- many baselines, especially commercial ones, only support their proprietary, high-quality voices. We refer to baselines that can support generating in the same speaker voices as ours as 'consistent speakers', and those that cannot as 'inconsistent speakers'. This was introduced in lines 290-291.
> >
> > The reason we separated out two sets of experiments for consistent speakers and inconsistent speakers is that subjective evaluation is very sensitive to speaker differences. Many listeners may like an utterance simply because they like the speaker's voice. To minimize the interference, we separate out the inconsistent speaker settings from the consistent ones.
> >
> > To make this more clear, we will change lines 290-291 into 'We conducted two sets of listening tests: 'Consistent speakers', which includes all the baselines that could synthesize the same speakers as ours, and 'Inconsistent speakers', including baselines that cannot. This could minimize the influence of speaker differences on perceived quality.'
> >
> > ### 5. 'Neutral' class excluded from emotion recognition experiment.
> >
> > We excluded the 'neutral' class because it is a special, 'emotionless' emotion. Recall that we need to turn the emotion classification task into an audiobook setting, and it is a bit odd to say 'He/She was neutral'. Nevertheless, we follow your advice and add the neutral sentiment back to the data. The revised results are as follows.
> >
> > | Methods | Happy | Sad | Excited | Angry | Neutral |
> > |---|---|---|---|---|---|
> > | **MIMI-tok** | -0.010 (0.005) | 0.154 (0.009) | 0.008 (0.006) | -0.013 (0.006) | -0.009 (0.007) |
> > | **GLM4V-tok** | -0.014 (0.005) | 0.030 (0.007) | -0.010 (0.005) | 0.002 (0.005) | 0.001 (0.006) |
> > | **ProsodyLM** | **0.014** (0.007) | **0.203** (0.015) | **0.203** (0.017) | **0.021** (0.010) | **0.063** (0.014) |
> >
> > As can be observed, the results are largely consistent with Table 3 in the main paper.

---

> > ### Author Response · Authors · 2025-06-02
> > **Response to Reviewer HdRD (Part 3)**
> >
> > ### 6. Would prosody tokens affect performance on semantically-rich tasks?
> >
> > Applying ProsodyLM to the intent classification (IC) is a non-trivial task, because recall that ProsodyLM was continuously pre-trained on audiobooks only, and thus does not have any instruction-following capabilities. This also makes comparison with a text-based instruction model quite an unfair comparison.
> >
> > While we are investigating how to adapt IC to an audiobook setting, we would like to present some alternative results. Specifically, we evaluate the text-section perplexity on some validation audiobooks (unseen during continuous pre-training), and compare it with two text-only LLM baselines: 1) Llama3.1-8b-instruct, and 2) Llama3.1-8b-instruct continuously fine-tuned on the same set of audiobooks. The results are as follows:
> >
> > | Llama3.1-8b-Instruct | Llama3.1-8b-Instruct Fine-Tuned | ProsodyLM |
> > |---|---|---|
> > | 24.53 | 11.7 | 13.87 |
> >
> > As can be observed, ProsodyLM only increases perplexity by 2 compared to pure text LLM pre-trained on the same data, which indicates that introducing prosody tokens has only minor effects on text modeling capabilities.
> >
> > ### 7. Speaker-dependent information
> >
> > Yes, the log-F0 for the speaker should be provided for both training and inference. The primary purpose of the log-F0 information is to distinguish male speakers from female speakers (males typically speak in lower F0s and females in higher). As a result, ProsodyLM is highly robust to fluctuations in the log-F0 values, as long as it still falls in a reasonable range for the speaker.
> >
> > In one-shot generation settings where we do not have a global statistics of the speaker, we will specify speaker log-F0 as the **average log-F0 of one example utterance of the speaker** (we need at least one utterance of a speaker to replicate the voice). In very rare cases where no example utterance is available, we will just specify it as bin 150 if the speaker is male and bin 300 if the speaker is female.
> >
> > Since log-F0 is a pretty generic information (and it is the only speaker-dependent information accessible to ProsodyLLM), ProsodyLLM would not 'memorize' certain speakers or certain speaker log-F0s. It works well for arbitrary unseen speakers.

---

> > > ### Author Response · Authors · 2025-06-02
> > > **Response to Reviewer HdRD (Part 4)**
> > >
> > > ### 8. Librilight transcriptions.
> > >
> > > We used Whisper transcriptions for the Librilight pre-training data.
> > >
> > > ### 9. Use cases for 'prosody->content' in intro.
> > >
> > > Thank you for your suggestions! We believe this case is actually much more suitable to mention in our section 1, so we will revise this section accordingly.
> > >
> > > ### 10. Misplaced appendix
> > >
> > > Thanks for catching this! We will move Appendix D4.1, Lines 704-711, to Appendix D4.2.
> > >
> > > ### Further comments on 'Reasons to Reject'
> > >
> > > Thank you for pointing out that some implementation details were missing in the main paper. Due to the loose template of COLM, we had to aggressively trim the main text and move many details to the appendix (for example, Appendix C was in the main text).
> > >
> > > To improve the clarity of our paper, we will make the following changes to our paper:
> > >
> > > * Move Appendix C back to Section 3 (as new Section 3.3). This includes all the key pre-training details, speaker info conditioning, and the instructions used in the continuous pre-training.
> > > * More details on the metric design of 'Prosody->Content' experiments, together with the rationales (answer to your question 3)
> > > * An added paragraph in Section 4 (before Section 4.1) explaining the details of synthesizing in different voices (answer to your question 1), and that some baselines can synthesize the same voices as ours, while others cannot.
> > > * Revising confusing sentences (line 243, lines 290-291)
> > > * Adding all aforementioned new experiments
> > >
> > > We hope these changes can significantly improve the clarity of the paper.

---

> > ### Author Response · Authors · 2025-06-07
> > **Intent Classification Experiment**
> >
> > We would like to follow up regarding the intent classification (IC) experiment (your question 6). We evaluate our method on the test set of [fluent_speech_commands_dataset](https://www.kaggle.com/datasets/tommyngx/fluent-speech-corpus). Recall that since ProsodyLM is not instruction-tuned, we need to modify the task as a next token prediction task (like Sections 4.3.1 and 4.3.2). Specifically, we adopt the following template for the three aspects of IC, action, object, and location, respectively:
> > ```
> > 'Turn off the lights.' [Prosody] Based on what the user said, the intention is to ___
> > 'Turn off the lights.' [Prosody] Based on what the user said, the object is ___
> > 'Turn off the lights.' [Prosody] Based on what the user said, the location is ___
> > ```
> > We then probe the ProsodyLM's output probabilities for a list of candidate continuations and see if the correct answer has the highest probability.
> >
> > We include two baselines. 1) Llama3.1-8b-instruct, and 2) Llama3.1-8b-instruct continuously fine-tuned on the same set of audiobooks as ProsodyLM. The second baseline is almost the same as ProsodyLM except that the [Prosody] section is not present during fine-tuning, so it can best demonstrate the effect of the [Prosody] section on this task.
> >
> > The following table shows the results.
> > | | Llama3.1-8b-Instruct | Llama3.1-8b-Instruct Fine-Tuned | ProsodyLM |
> > |---|---|---|---|
> > | **Action** | 91.15 | 87.02 | 87.88 |
> > | **Object** | 87.35 | 86.26 | 93.62 |
> > | **Location** | 94.81 | 95.02 | 87.91 |
> >
> > As can be observed, ProsodyLM does not significantly underperform text-based LLMs, which implies introducing the [Prosody] section does not significantly impact the performance even on this task. Note that the results are not directly comparable with the reported results in existing literature because the former are adapted to the audiobook completion setting.
> >
> > Meanwhile, since it is approaching the deadline of the discussion period, but no discussion has been initiated so far, we would like to follow up on whether your concerns can be adequately addressed. We are happy to engage in discussion with you and answer any further questions you might have. Thank you for your time!

---

> > > ### Comment · Reviewer_HdRD · 2025-06-09
> > >
> > > Thanks for providing these additional results on IC. May I suggest including the macro/micro average over all classes in the revised version of the paper?

---

> > > > ### Author Response · Authors · 2025-06-09
> > > >
> > > > Dear Reviewer HdRD,
> > > >
> > > > Thank you for your feedback! We will make the suggested changes and report macro/micro averages for the IC experiments, along with other changes & experiments mentioned in our previous response.
> > > >
> > > > Thank you!
> > > >
> > > > The authors

---

> > ### Comment · Reviewer_HdRD · 2025-06-09
> >
> > Dear authors, thank you for the answers.
> >
> > Regarding (4): The use of the word "speakers" suggests that the samples were spoken by human speakers. To avoid this misunderstanding, I would suggest replacing "consistent/inconsistent speakers" with "consistent/inconsistent voices".
> >
> > Regarding (6): The results on perplexity are interesting. However, it is worth noting that perplexity does not always correlate with downstream task performance, see e.g.  Levy et al., 2024; Liu et al., 2023:
> >
> > Hong Liu, Sang Michael Xie, Zhiyuan Li, Tengyu Ma. "Same Pre-training Loss, Better Downstream: Implicit Bias Matters for Language Models", Proceedings of the 40th International Conference on Machine Learning, PMLR 202:22188-22214, 2023.
> >
> > Levy, Mosh, Alon Jacoby, and Yoav Goldberg. “Same Task, More Tokens: The Impact of Input Length on the Reasoning Performance of Large Language Models.” In _Proceedings of the ACL (Volume 1: Long Papers), 2024.

---

> ### Author Response · Authors · 2025-06-09
> **Follow up on Rebuttal Discussion**
>
> Dear Reviewer HdRD,
>
> We’re writing again regarding your review and our response. With only two days remaining in the discussion period, we would greatly appreciate hearing whether our revisions have addressed your concerns.
>
> We dedicated substantial effort to responding to your comments. Specifically:
>
> * Regarding your concern about **clarity**, we have committed to reintegrating key implementation details into the main paper, along with additional clarifications.
>
> * Regarding **experimental design**, we have included new experiments and clarified our evaluation metrics.
>
> We believe these revisions significantly strengthen our submission. Your follow-up would be valuable in ensuring a fair assessment of the paper. Please let us know if you have any additional questions or feedback. We are happy to engage further.
>
> Thank you very much for your time!
>
> Sincerely,
>
> The Authors

---

> > ### Comment · Reviewer_HdRD · 2025-06-09
> >
> > Dear authors, thank you for the detailed answers. I have changed my rating accordingly.

---

### Official Review · Reviewer_mFRG · 2025-05-13

**Rating:** 7
**Confidence:** 4
**Ethics Flag:** 1

**Summary:**

This paper introduces ProsodyLM which introduces a simple prosody tokenization scheme and demonstrates that compared with conventional speech tokenization schemes, the proposed tokenization scheme can learn diverse prosody processing capabilities through pre-training alone and shows promising results.

The main novelty of this paper is the new prosody token and the way authors formulate and breakdown the problem of Prosody in output speech. The paper breaks down the prosodic dependencies into three categories -
Content -> Prosody
Prosody -> Content
Prosody -> Prosody

The paper does a thorough analysis of the proposed system and compares it to various baselines. The paper also shares the audio evaluation samples.

**Reasons To Accept:**

* The main ideas of the paper are simple and intuitive
* The evaluation is thorough

**Reasons To Reject:**

* Would have loved to see the feature analysis of the new prosody token
* Typo/Grammar: Line 171: Speech “Encoer”

---

> ### Author Response · Authors · 2025-06-02
> **Response to Reviewer mFRG**
>
> Thank you for the positive feedback! Following your suggestion on **feature analysis**, the following links contain histograms of the prosody feature bins for each of the five prosody features, [F0 median](https://raw.githubusercontent.com/anonymous0818/anonymous0818.github.io/refs/heads/main/rebuttal/hist/f0_median.png), [F0 range](https://raw.githubusercontent.com/anonymous0818/anonymous0818.github.io/refs/heads/main/rebuttal/hist/f0_range.png), [F0 slope](https://raw.githubusercontent.com/anonymous0818/anonymous0818.github.io/refs/heads/main/rebuttal/hist/f0_slope.png), [energy](https://raw.githubusercontent.com/anonymous0818/anonymous0818.github.io/refs/heads/main/rebuttal/hist/energy.png), and [duration](https://raw.githubusercontent.com/anonymous0818/anonymous0818.github.io/refs/heads/main/rebuttal/hist/duration.png), respectively. As can be observed, F0 median has a bi-modal distribution, each of the two modes representing male and female speakers, respectively. F0 range and energy have pretty skewed distributions, while duration and F0 slope are relatively centered. We will add these results to the paper.

---

### Decision · Program_Chairs · 2025-07-08

**Decision:**

Accept

**Comment:**

The paper presents a successful augmentation of a language-audio model that novelly explicitly represents extracted prosodic features.  Given the discussion between authors and reviewers, it seems that the final paper will be complete (thorough evaluation and analysis) and well written.

Main pro:
-the paper recognises the importance of rich representation of prosody in speech models and proposes a successful solution

Main con:
-the augmentation is rather small, though impactful